# Decoding the epigenetics and chromatin loop dynamics of androgen receptor-mediated transcription

Umut Berkay Altıntaş [1,13], Ji-Heui Seo[2,13], Claudia Giambartolomei [3,4], Dogancan Ozturan[1], Brad J. Fortunato[2], Geoffrey M. Nelson [5,6], Seth R. Goldman [5], Karen Adelman [5,7], Faraz Hach[1,8,9], Matthew L. Freedman [2,7,10,14] & Nathan A. Lack [1,8,11,12,14] ✉

Androgen receptor (AR)-mediated transcription plays a critical role in development and prostate cancer growth. AR drives gene expression by binding to thousands of cis-regulatory elements (CRE) that loop to hundreds of target promoters. With multiple CREs interacting with a single promoter, it remains unclear how individual AR bound CREs contribute to gene expression. To characterize the involvement of these CREs, we investigate the AR-driven epigenetic and chromosomal chromatin looping changes by generating a kinetic multi-omic dataset comprised of steady-state mRNA, chromatin accessibility, transcription factor binding, histone modifications, chromatin looping, and nascent RNA. Using an integrated regulatory network, we find that AR binding induces sequential changes in the epigenetic features at CREs, independent of gene expression. Further, we show that binding of AR does not result in a substantial rewiring of chromatin loops, but instead increases the contact frequency of pre-existing loops to target promoters. Our results show that gene expression strongly correlates to the changes in contact frequency. We then propose and experimentally validate an unbalanced multi-enhancer model where the impact on gene expression of AR-bound enhancers is heterogeneous, and is proportional to their contact frequency with target gene promoters. Overall, these findings provide insights into AR-mediated gene expression upon acute androgen simulation and develop a mechanistic framework to investigate nuclear receptor mediated perturbations.

The androgen receptor (AR) is a ligand-dependent transcription factor that plays a critical role in regulating gene expression in the prostate[1]. In its inactive form, the AR resides in the cytoplasm where it is stabilized by heat-shock chaperone proteins[2,3]. After binding androgens, such as testosterone or dihydrotestosterone (DHT), the AR undergoes an allosteric modification and translocates into the nucleus[2–4]. Once there, the AR binds to specific cis-regulatory elements (CREs) on DNA through an interplay of chromatin accessibility, pioneer factors such as

FOXA1, and sequence motifs[5–9]. The majority of these AR-bound CREs are proposed to function as enhancers as they are both located distal from gene promoters[10] and are brought into close physical proximity by chromatin loops[11,12]. The enhancer activity at these CREs is associated with various epigenetic features, including chromatin accessibility, transcription factor binding, and post-translational histone modifications, such as H3K27ac[13–19]. CREs are proposed to impact transcription through chromatin contacts with the target promoter

---

that cause the recruitment of co-regulatory proteins and transcriptional machinery[20–23]. This typically involves multiple AR-bound enhancers and CREs that interact with the target promoter[24–30]. The contribution of individual CREs has been controversial, with some studies suggesting that CREs work additively to increase gene expression[31–33], while others propose that many enhancers are redundant and only contribute at specific developmental stages[34–37]. The complex kinetic interplay of epigenetic modifications, co-regulatory proteins and chromatin loops across multiple CREs following transcription factor binding is poorly understood.

To explore how AR binding impacts epigenetic modifications and chromatin looping at regulatory elements in response to an acute perturbation, we generated an extensive multi-omics experimental dataset following androgen stimulation that is integrated into a graph-based framework. We demonstrated that AR binding sequentially induces an increase in FOXA1 and H3K27ac signals, that is followed by an increase in chromatin accessibility. We further show that AR does not induce new chromatin loops, but instead increases the contact frequency between gene promoters and selective AR-bound enhancers. From these results, we proposed and validated a multi-enhancer model, where a small subset of pre-established dominant CREs with increased chromatin contact frequency exhibits an elevated dynamic response to androgen stimulation, which significantly contributes to gene expression. These results provide a foundation for understanding how enhancers respond to an acute perturbation.

## Results

### Generation of androgen-stimulation kinetic dataset and construction of regulatory networks

To characterize the temporal impact of AR binding on epigenetic features and chromatin looping, we generated an extensive kinetic multi-omics experimental dataset following androgen stimulation. We treated LNCaP cells with androgen (dihydrotestosterone) and collected cells at 5 different time points (0 m, 30 m, 4 h, 16 h, 72 h). At each time point, multiple features were characterized, including gene expression (RNA-seq), chromatin accessibility (ATAC-seq), transcription factor binding, and post-translational histone modification (AR, FOXA1, H3K27ac, and H3K4me3 ChIP-seq), chromatin looping (HiChIP) and capped nascent RNA (Start-seq) (Fig. 1A). From these datasets, CREs (n = 78,522) were defined from accessible sites (ATAC-seq)[15]. Based on known gene annotations and AR ChIP-seq, CREs were annotated as either promoters (n = 13,452), AR-free CREs (n = 59,570), or AR-bound CREs (n = 5231). The interaction between these CREs was defined from consensus H3K27ac (enhancer-centric; n = 296,326) and H3K4me3 (promoter-centric; n = 278,491; see methods) HiChIP loop sets. Demonstrating that these histone marks

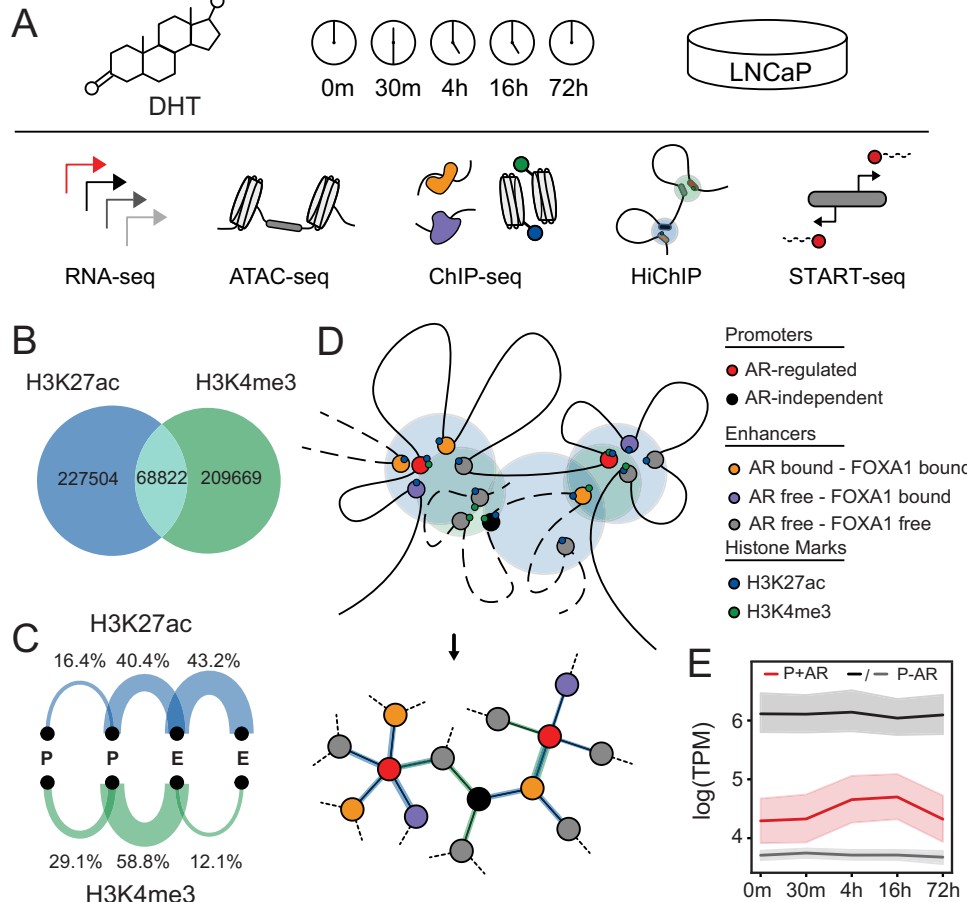

**Fig. 1 | Design of temporal multi-omics dataset and construction of a bioinformatic framework. A** Schematic representation of the experimental design. LNCaP cells were treated with 10 nM DHT and samples were collected at five different time points (0 m, 30 m, 4 h, 16 h, 72 h) for RNA-seq, ATAC-seq, ChIP-seq (AR, FOXA1, H3K27ac, H3K4me3), HiChIP (H3K27ac, H3K4me3), and Start-seq. **B** Venn diagram representing significantly called chromatin loops from merged H3K27ac and H3K4me3 HiChIP datasets. **C** Arc plots representing percentages of promoter-promoter (P-P), enhancer/CRE-promoter (E-P), and enhancer/CRE-enhancer/CRE (E-E) loops for H3K27ac and H3K4me3 HiChIP. **D** Graphical representation of a regulatory network. **E** Gene expression profile of androgen response hallmark genes (P + AR; red), highly expressed (first quartile; P-AR; black) and mid-high (second quartile; P-AR; gray) expressed genes at all time points. The solid line represents the mean, and the error bars indicate the 95% confidence.

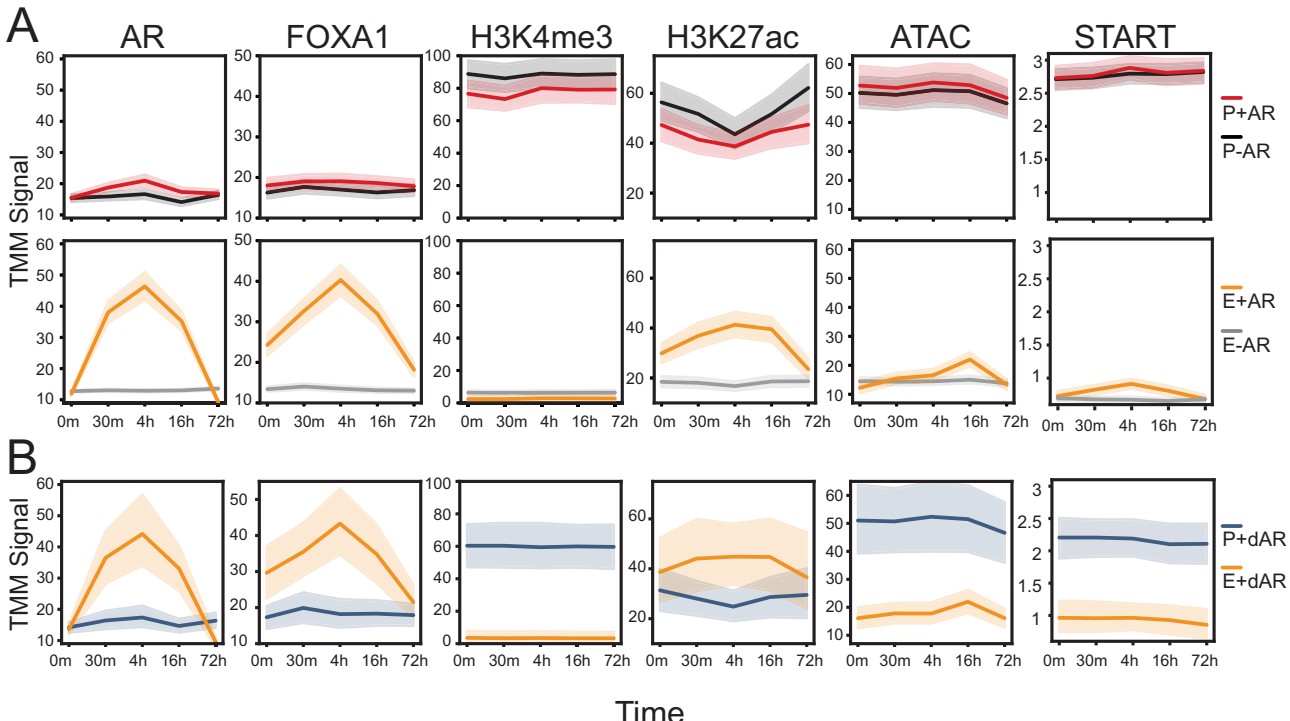

**Fig. 2 | Activation of the androgen receptor leads to a delayed increase in histone modifications and chromatin accessibility. A** Trimmed mean of M values (TMM) normalized ChIP-seq or ATAC-seq or Start-seq signal were compared across all time points at different regulatory elements, including: Promoters of AR upregulated genes (P + AR: red), promoters of AR-independent genes (P-AR: black), AR-bound enhancers (E + AR: yellow) of AR-regulated genes, AR-free enhancers (E-AR: gray) of AR-independent genes. **B** A similar analysis was done for the promoters of AR-downregulated genes (P+dAR: blue), and their AR-bound enhancers (E+dAR: yellow). For all figures, the solid line represents the mean, and the error bars indicate the 95% confidence.

are associated with different functional CREs, only 23% of loops were found in both H3K27ac and H3K4me3 HiChIP (Fig. 1B). As expected, H3K27ac loops were predominantly between enhancer-enhancer (E-E) pairs, accounting for 43.2% of the loops, followed by enhancer-promoter (E-P) pairs, which constituted around 40.4% (Fig. 1C). In contrast, H3K4me3 loops were primarily between E-P pairs, making up 58.8% of the total loops. This is consistent with the known associations of these histone marks to promoter and enhancer CREs[38–40]. To allow more quantitative analyses of these large-scale chromatin interactions, we transformed the resulting HiChIP looping data into a graphical network with each node representing a CRE and the edges being the chromatin loops between these two elements (Fig. 1D). With this regulatory network, we then overlaid the various multi-omics datasets to provide a framework that can interrogate the impact of AR binding. We investigated androgen-driven AR-mediated gene transcription based on the previously characterized hallmark androgen-responsive genes and observed strong upregulation following androgen treatment (Supplementary Fig. 1A, B). Expression heatmaps indicate that these previously identified androgen-responsive genes predominantly exhibit upregulation upon androgen treatment, with peak expression occurring at distinct time points (Supplementary Fig. 1C, D). As expected, these genes were significantly induced compared to similarly expressed random AR-independent genes (Fig. 1E). This comprehensive multi-omics dataset and graphical regulatory network provided a bioinformatic framework to quantitatively investigate the temporal impact of AR binding on epigenetic features, chromatin looping, and gene expression.

## Androgen stimulation leads to stepwise epigenetic changes
With this structured regulatory network, we then explored how AR binding affects epigenetic features and their relationship with gene

transcription. The AR-regulated genes' promoters (P + AR) and their looped AR-bound enhancers (E + AR) were compared to random highly expressed AR-independent genes' promoters (P-AR) and their looped AR-free enhancer (E-AR) (Fig. 2A). We observed that E + AR displayed a strong AR and FOXA1 signal following androgen stimulation, which reached its peak at 4 h and then subsequently decreased at 16 and 72 h. Emphasizing FOXA1's pioneering activity, E + AR displayed an elevated FOXA1 signal at the initial time point (0 m). As expected, the AR and FOXA1 signals did not significantly change at AR-independent promoters or AR-free enhancers. Interestingly, those FOXA1-bound CREs that were not co-occupied by AR had no change in the relative FOXA1 signal (Supplementary Fig. 2A, C), suggesting that AR potentially influences FOXA1 occupancy[41]. We observed a higher H3K4me3 ChIP-seq signal at promoters compared to enhancers. This signal was largely unaffected by AR binding, but there were selective genes, including *KLK3*, which exhibited an increasing H3K4me3 mark at its promoter following androgen treatment (Supplementary Fig. 3). We observed an elevated H3K27ac signal at promoters compared to enhancers (Fig. 2A). Further, the H3K27ac signal increased specifically at E + AR, while it remained unchanged at E-AR. This change at E + AR was also observed for chromatin accessibility (ATAC-seq), though the maximum signal (16 h) was found to occur after AR and FOXA1 peak occupancy (4 h). We also investigated the epigenetic changes of AR downregulated gene promoters and their AR-bound enhancers (Supplementary Fig. 4). Interestingly, we observed that the AR-bound CREs (E+dAR) looped to the promoter of downregulated genes (P+dAR) displayed a very similar increase in AR, FOXA1, H3K27ac, and chromatin accessibility (Fig. 2B). There was no statistically significant change at any time point in the AR-bound enhancers of either up or downregulated genes ($p > 0.1$). However, nascent enhancer RNA (eRNA) transcription at AR-bound enhancers looped to AR-upregulated gene promoters (E + AR) behaved differently than those

AR-bound enhancers looped to promoters of AR-downregulated genes (E + dAR). The eRNA transcription at AR-bound enhancers looping to AR-upregulated genes had similar dynamics to AR-binding with ChIP-seq peaking at 4 h. In contrast AR-downregulated loops did not exhibit such dynamics, despite being bound by AR. Notably, this peak in eRNA transcription at AR-bound enhancers preceded changes in accessibility, indicating a parallel layer of regulation. Overall, these kinetic datasets show that there is a sequential process that occurs following androgen stimulation where AR, FOXA1, eRNA, and H3K27ac signals selectively increase at looped enhancers, before inducing subsequent changes in chromatin accessibility.

### AR-bound enhancers increase contact frequency to AR-upregulated gene promoters

Next we investigated how chromatin looping changed following AR activation. Initially, we compared the number of loops formed following androgen stimulation and found that both promoters of AR-upregulated genes (P + AR) and their looped AR-bound enhancers (E + AR) did not exhibit any significant changes ($p > 0.05$) in the number of loops compared to background P-AR/E-AR (Fig. 3A). There was also no significant difference in the distribution of the number of loops on gene promoters during androgen treatment (Supplementary Fig. 5A). These results demonstrate that AR binding does not cause a substantial rewiring of chromatin looping.

While androgen treatment did not significantly change the number of loops, we did observe an increase in contact frequency at AR-bound enhancers looping to AR-regulated genes (Supplementary Fig. 6). To quantify these changes, we calculated the fold change in contact frequency compared to bootstrapped ($b = 1000$) random AR-independent genes from both promoter and enhancer viewpoints (Fig. 3B). We observed that AR activation increased the contact frequency of loops at AR-regulated promoters (P + AR) over time, with a peak at 16 h (H3K27ac: $p < 0.001$; H3K4me3: $p < 0.001$) (Fig. 3C). In contrast, the change in contact frequency of loops to promoters of AR-independent genes (P-AR) remained stable. From an enhancer viewpoint (Fig. 3D), AR-bound enhancer CREs (E + AR) showed a significant increase in both H3K27ac and H3K4me3 contact frequency (H3K27ac: $p < 0.001$; H3K4me3: $p < 0.001$). In contrast, AR-free CREs (E-AR; gray) that were connected to the same AR-regulated gene promoters did not exhibit any significant change. Further, no change in contact frequency was observed in the CREs (E-ARi; black) that interact with AR-independent gene promoters. While chromatin contact frequency increased between AR-bound enhancer CREs (E + AR) and upregulated hallmark gene promoters (P + AR), we did not observe any change in loops between AR-bound CREs (E + dAR) and downregulated gene promoters (Fig. 3E). This is particularly striking as, AR-bound CREs (E + AR, E+dAR) exhibited a similar pattern in their epigenetic profiles, regardless of whether they looped to an upregulated or downregulated gene (Fig. 2A, B). Given that a similar trend is observed at all AR-bound CREs, this suggests that epigenetic modifications alone do not determine gene expression and combine with chromatin looping.

To interrogate the kinetic changes in epigenetics and contact frequency from multiple CREs, we focused on several upregulated androgen-responsive genes (KLK2, KLK3, NKX3-1, UAP1, ABCC4, and DHCR24) (Fig. 3F). As expected, AR-free enhancers (E-AR; gray) had minimal changes in epigenetics and contact frequency. Interestingly, we observed that AR-bound enhancers (E + AR; orange) had broad heterogeneity in response to treatment and the same enhancer typically had both the highest change in epigenetic features and contact frequency. Our findings demonstrate that AR-bound enhancers increase contact frequency with AR-regulated gene promoters in response to androgen treatment. However, this response is heterogeneous, and there is significant variability among AR-bound enhancers.

### Association of nascent transcription to epigenetic changes and contact frequency

To determine if the change in contact frequency precedes or occurs simultaneously with AR-mediated gene transcription, we characterized the kinetic rate of androgen-induced gene expression. We identified upregulated androgen-induced genes from nascent RNA (Start-seq) and categorized them based on the maximal expression (30 m, 4 h, 16 h, or 72 h) (Fig. 4A). We chose to group these genes based on nascent RNA, as RNA-seq provides only steady-state quantification of mRNA transcripts[42]. Between these groups, no significant difference was observed in the maximal signal at AR-bound enhancers for AR, FOXA1, and H3K27ac ChIP-seq or chromatin accessibility (Fig. 4B), suggesting that the epigenetic features do not dictate the timing of nascent RNA production. However, we observed a temporal relationship between nascent transcription and chromatin contact frequency. Specifically, the maximal nascent transcription (4 and 16 h groups) occurred simultaneously with an increase in H3K27ac and came before the peak of H3K4me3 (16 h) contact frequency at AR-bound enhancers (Fig. 4C). These observations indicate that the change in contact frequency does not precede RNA transcription. Overall, these findings suggest that although AR and FOXA1 rapidly bind at enhancers upon androgen treatment, the maximal transcription occurs either simultaneously with or before a maximal alteration in chromatin contact frequency.

### Multi-enhancers influence transcription proportional to contact frequency

Having observed marked heterogeneity in chromatin loop contact frequency at AR-bound enhancers, we began to explore the impact of multi-enhancer contacts on gene expression. We expanded the scale of our analysis to include all detectable genes ($n = 13,575$) and characterize how loop frequency correlates to gene expression. Most genes had multiple CREs interacting with target promoters with a median frequency of ~15 interactions per gene (Supplementary Fig. 8A). With this large dataset, we evaluated three models correlating H3K27ac and H3K4me3 contact frequency to gene expression where: all interactions contribute equally to gene expression (average), only a single strong interaction affects gene expression (maximum), or all interactions contribute to gene expression in an additive manner (sum) (Fig. 5A). While all models strongly correlated H3K27ac and H3K4me3 contact frequency to gene expression ($p < 0.0001$; Spearman's test), we found that the sum model (H3K27ac; $r = 0.896$, H3K4me3; $r = 0.904$) and maximum model (H3K27ac; $r = 0.839$, H3K4me3; $r = 0.876$) correlated significantly better than the average model (H3K27ac; $r = 0.643$, H3K4me3; $r = 0.679$) across all time points (Fig. 5B and Supplementary Fig. 9). To quantify the importance of each feature, we built a random forest regressor to predict the expression from these operations (Accuracy; $r^2 = 0.805$) and found that the additive model (sum) best-predicted expression (Supplementary Fig. 8B)[43]. As we observed a similar correlation with the maximum model, which scored only a single chromatin loop, this suggested that the contact frequency of chromatin loops to a promoter were markedly unbalanced and that there was a "dominant" interaction. To characterize this behavior within the gene context, we quantified the inequality of chromatin loop contact frequency to a target promoter and found that both H3K27ac and H3K4me3 were strongly unbalanced (Gini >~0.5; Supplementary Fig. 8C). The length distributions of accessible sites (ATAC-seq peaks) within dominant and non-dominant interactions were not different (Supplementary Fig. 8D). Our attempts to identify known DNA motif sequences associated with dominant AR-bound enhancers did not yield promising results. These dominant loops did not relate to prostate cancer risk variants (Fisher's exact test $p > 0.05$) (Supplementary Data 4). These findings suggest that enhancers interacting with gene promoters do not have a uniform distribution in

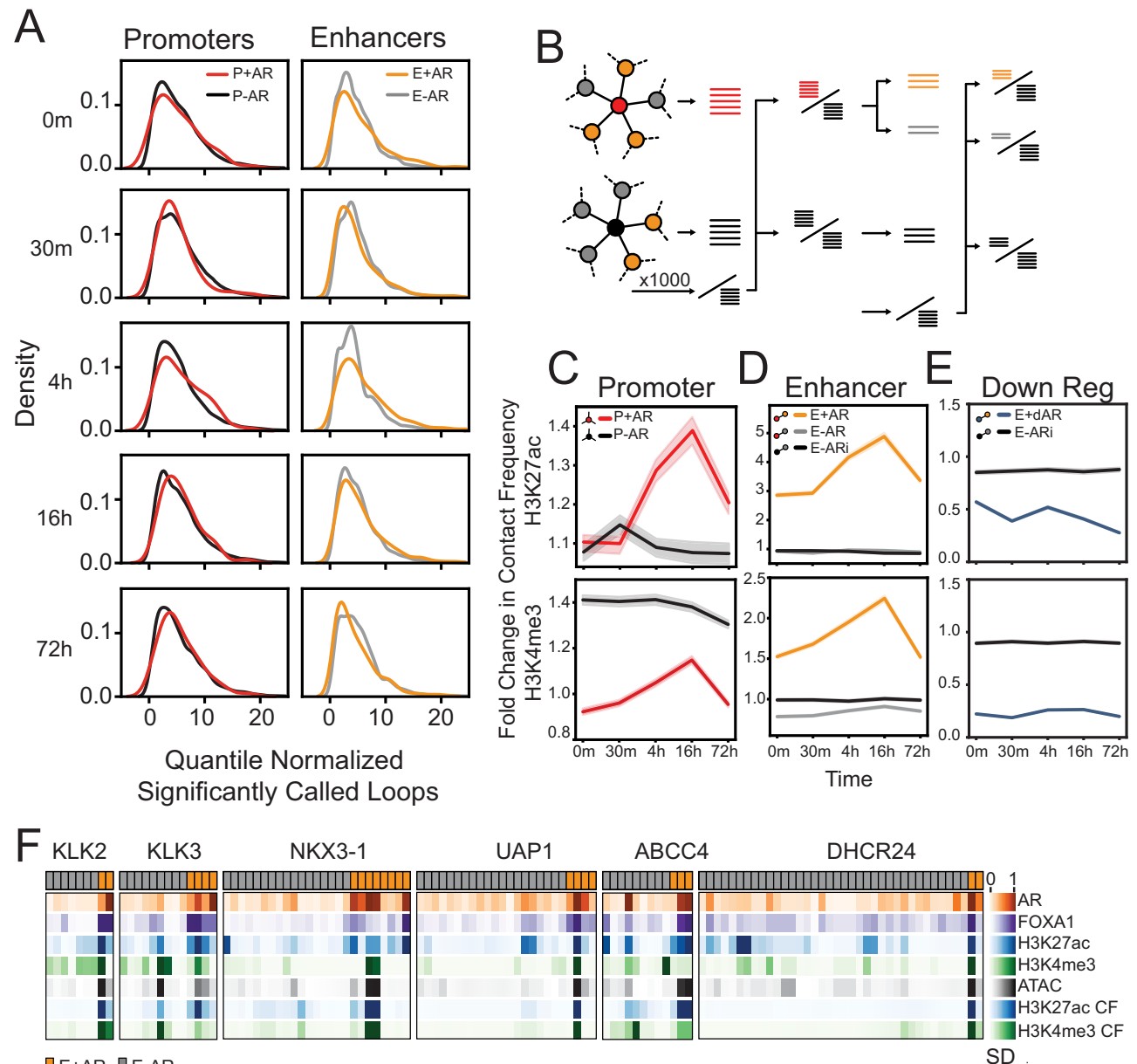

**Fig. 3 | AR-bound enhancers only increase contact frequency to AR-regulated gene promoters. A** Kernel density estimation of the number of significantly called loop anchoring promoters (left; AR-regulated promoters are in red; AR-independent promoter backgrounds are in black) or enhancers (right; AR-bound enhancers are in yellow; AR-free CRE background are in gray) at each time point (H3K27ac HiChIP). Each row represents the significantly called loops at each time point separately, and not reference loops. **B** Schematic representation of fold change in contact frequency calculation for a given gene set (left), from its promoter's viewpoint (middle), from its enhancers' viewpoint (right). The query loop sets are compared to the same reference loop set. Promoter view: The AR-regulated gene promoter loops (red sticks) and a random set of AR-independent gene promoter loops (black sticks) are compared to the reference loops, which are selected from AR-independent promoter loops (denominator black sticks). Enhancer view: The loops of AR-bound enhancers (yellow sticks) and AR-free enhancers (gray sticks), and loops of AR-free enhancers (black sticks) that interact with AR-independent gene promoters are compared to the reference loops which are

selected from AR-independent promoters that interact with AR-free enhancers (denominator black sticks). **C** Fold change in chromatin loop contact frequency of AR-regulated gene promoters (P + AR; red) and highly expressed AR-independent gene promoters (P-AR; black). **D** Fold change in contact frequency of AR-bound enhancers that loop to AR-regulated genes (E + AR; yellow), AR-free enhancers looping to AR-regulated genes (E-AR; gray), and AR-free enhancers looping to highly expressed AR-independent genes (E-ARi; black). **E** Fold change in contact frequency in AR-bound enhancers looping to AR-downregulated genes (E+dAR; blue), and AR-free enhancers looping to highly expressed genes (E-ARi; black). **F** Min-max normalized standard deviation (SD) following androgen treatment for AR, FOXA1, H3K27ac, and H3K4me3 ChIP-seq, ATAC-seq and H3K27ac and H3K4me3 HiChIP contact frequency (CF) are depicted for AR-bound (E + AR; orange) and AR-free (E-AR; gray) CREs of six selected AR-regulated genes (*KLK2, KLK3, NKX3-1, UAP1, ABBCC4,* and *DHCR24*). For all line plots (**C**−**E**), the solid line represents the mean, and the error bars indicate the 95% confidence.

contact frequency and that there are "dominant" loops that strongly correlate with expression (Fig. 5D).

To better understand how these potential dominant loops are dynamically affected by acute androgen perturbation, we

characterized the CRE-promoter interactions of androgen-regulated genes (*n* = 88; Supplementary Data 3). Dominant loops were identified for every gene promoter by first scaling the contact frequency of interacting CREs (0, 1 range), and selecting the highest ones with a

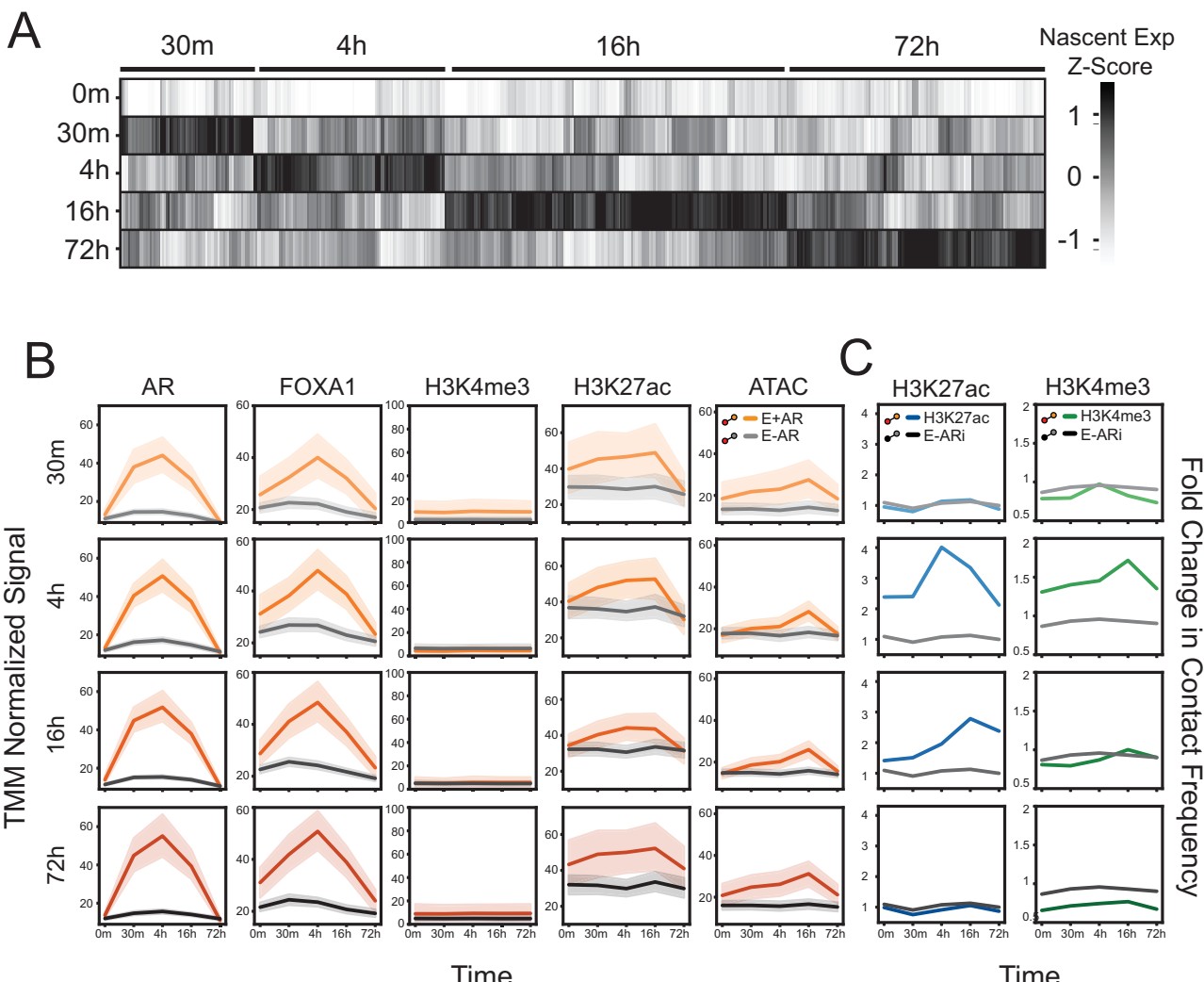

**Fig. 4 | Temporal changes of nascent RNA. A** Clustering of nascent capped mRNA (Start-seq) according to time-dependent maximum expression. The union of upregulated transcripts' nascent expression level across all time points were z-score normalized. **B** TMM normalized ChIP-seq or ATAC-seq of the following regulatory elements: AR-bound enhancers (E + AR: yellow) and AR-free enhancers (E-AR: gray). Each row represents the enhancers that loop to those genes that are maximally expressed at that time point (see **A**). **C** Fold change in contact frequency of AR-bound enhancers looping to maximally expressed genes for H3K27ac (E + AR; blue) and H3K4me3 (E + AR; green) HiChIP. These CREs were compared to AR-free enhancers of highly expressed genes (E-ARi; black) in both HiChIP datasets. For all line plots (**B**, **C**), the solid line represents the mean, and the error bars indicate the 95% confidence.

threshold >0.8. The AR dominant loops were not solely based on proximity to the gene promoter, as they were the closest CRE for only 40% of AR-regulated genes at any time point. The same dominant loops were commonly found before androgen treatment suggesting that the dominant loops are "primed" before AR binds (Supplementary Fig. 10). We found that the dominant AR-bound CREs (D+) had significantly ($p < 0.0001$) higher dynamic change in contact frequency than non-dominant AR-bound CREs (D-) that interacted with AR-regulated gene promoters (Fig. 5C). Interestingly, the dominant loops were highly gene-specific. When we characterized two AR-regulated genes (*KLK3* and *KLK2*) that share many looped CREs (Fig. 5E), we observed gene-specific changes in contact frequency changes from an AR enhancer (ARBS3) to the *KLK2* or *KLK3* promoters. Specifically, ARBS3 displayed the strongest and most dynamic contact frequency with the *KLK2* promoter, but not the *KLK3* promoter. Instead, the dominant *KLK3* promoter loop was with the well-known upstream AR-enhancer[44] (ARBS2). Interestingly, both these dominant looped enhancers were not the CRE with the highest change in AR peak height, suggesting that there are additional factors that contribute to changes

in contact frequency. Overall, these results demonstrate that AR binding does not affect chromatin looping equally and that those CREs with the most dynamic contact frequency potentially have a greater impact on gene transcription.

## CRISPR-based perturbations confirm the existence of "dominant" chromatin loops

To experimentally validate these correlative models, we tested the effects of dominant chromatin loops on androgen-induced gene expression. Utilizing CRISPRi, a derivative of the CRISPR/Cas9 system that inhibits enhancer activity without altering DNA sequences, we targeted all AR-bound enhancers ($n = 20$) that interact with (Supplementary Data 1, 2 and Supplementary Fig. 11) the previously described AR-regulated genes (*KLK2, KLK3, NKX3-1, UAP1, ABCC4,* and *DHCR24*). Across all tested genes, we found that inhibiting those CREs that had a dominant chromatin loop, significantly impacted androgen-induced transcription (Fig. 6A). For *KLK2* and *KLK3* we observed the highest inhibition of androgen-induced expression when we inhibited the AR-bound CREs that had the largest change in contact frequency (ARBS3

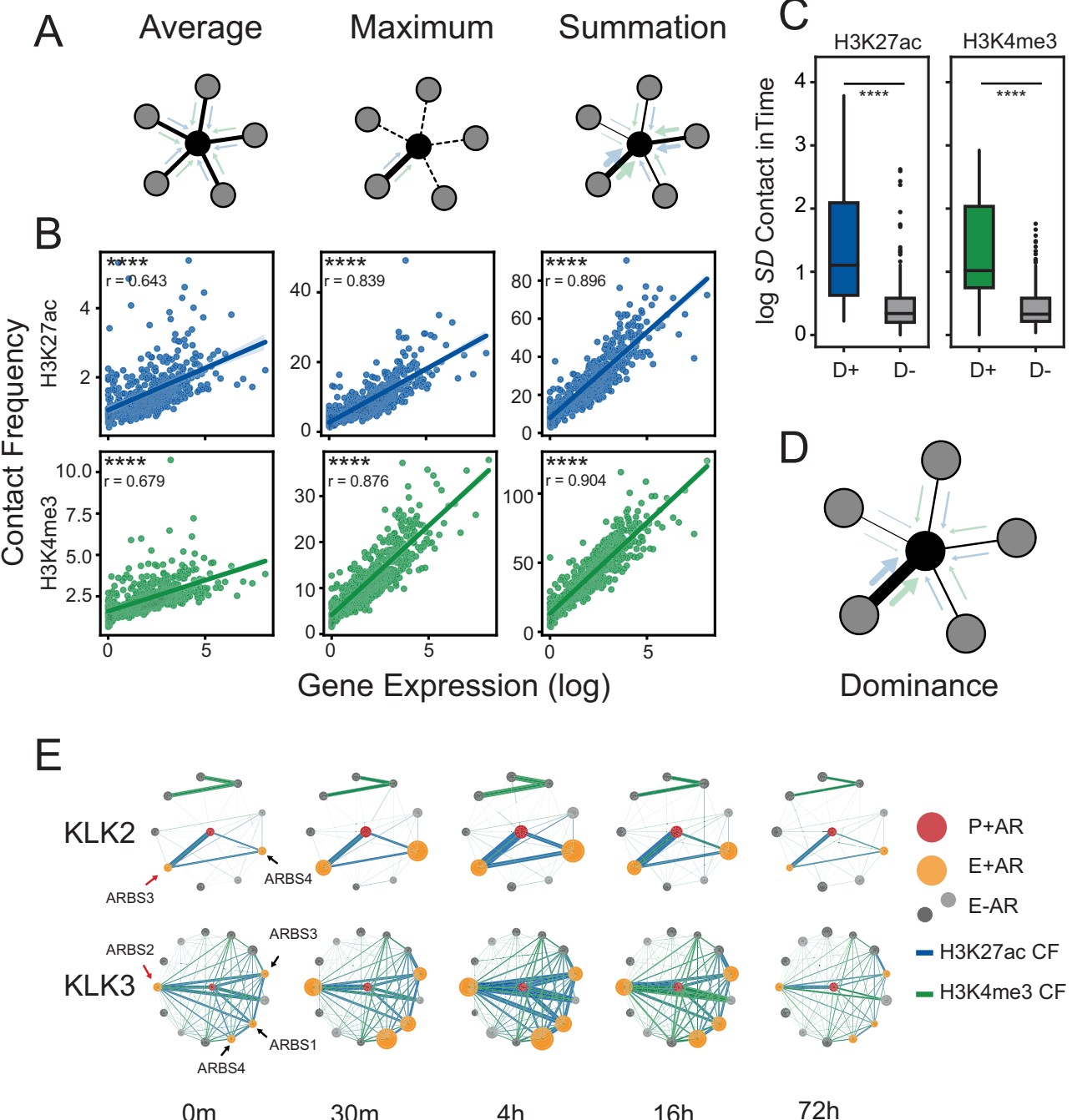

**Fig. 5 | Multi-enhancer contact is strongly influenced by a dominant loop.**
**A** Schematic representation of tested multi-enhancer models. The average model (left) represents an equal impact of all CREs (surrounding nodes; gray) on gene promoter (center node; black). The maximum model (middle) represents only one neighbor (unbroken line) impact on the gene promoter (center node; black). The summation model (right) proposes that all variable CREs (different widths of blue/green arrows) impact additively with their contact frequency to gene promoter.
**B** Scatter plot of binned (k = 25) log expression (x-axis) vs. contact frequency (y-axis) according to the proposed multi-enhancer models function (16 h HiChIP). The solid line represents the mean, and the error bars indicate the 95% confidence. The correlation was calculated by linear regression. The significance was assessed by Spearman's rank correlation ($p_{H3K27ac\_avg}$ -1.81 × 10e-60, $p_{H3K27ac\_max}$ -3.07 × 10e-135, $p_{H3K27ac\_sum}$ -1.69 × 10e-179, $p_{H3K4me3\_avg}$ -1.42 × 10e-69, $p_{H3K4me3\_max}$ -1.74 × 10e-161,

$p_{H3K4me3\_sum}$ -3.46 × 10e-188). **C** Standard deviation of H3K27ac and H3K4me3 HiChIP contact frequency change over time for dominant AR-bound CREs (D+) and non-dominant AR-bound CREs (D−) that interact with promoters of AR-regulated genes. The significance was assessed by Mann–Whitney $U$-test ($p_{H3K27ac}$ -6.41 × 10e-12, $p_{H3K4me3}$ -4.34 × 10e-10). **D** Schematic representation of hypothesized multi-enhancer dominance model. The width of the arrows (blue/green) represents the contact frequency. **E** Circle plot representation of first-degree regulatory interactions of *KLK2* (top) and *KLK3* (bottom) genes. Promoters are shown in the center (red) and the first-degree interactions are either AR-bound (yellow) or AR-free (grays) CREs. The size of the nodes represents the AR ChIP-seq signal, and the width of the edges represents contact frequency (H3K27ac CF: blue, H3K4me3 CF: green). For all data ns $p > 0.05$, *$p < 0.05$, **$p < 0.01$, ***$p < 0.001$, ****$p < 0.0001$.

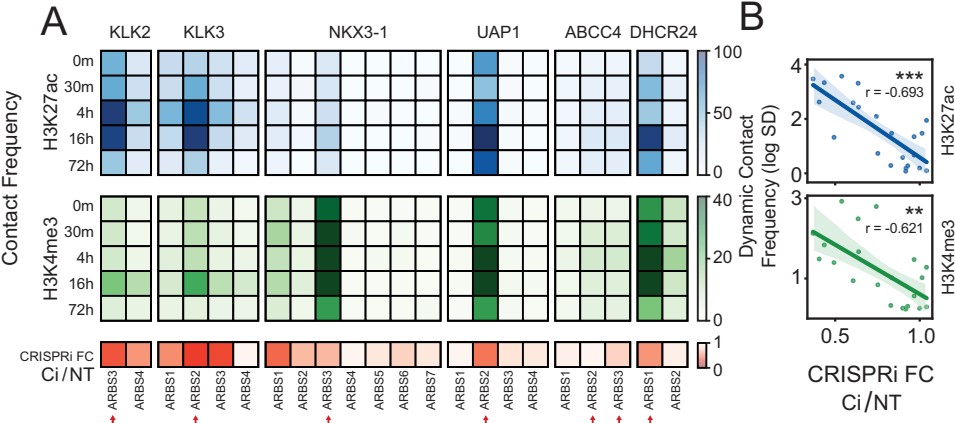

**Fig. 6 | Androgen-induced gene expression is significantly affected by perturbing the most dominant AR-bound enhancers. A** Functional characterization of individual AR-bound CRE on androgen-induced expression. Heatmap of contact frequency strength at each time point (H3K27ac: blue, H3K4me3: green) for individual AR-bound CREs to a target gene promoter (*KLK2*, *KLK3*, *NKX3-1*, *UAP1*, *ABCC4*, and *DHCR24*). Each AR-binding site (ARBS) was individually inhibited with CRISPRi and treated with androgen (10 nM DHT). The inhibition of the target gene induction was calculated compared to the non-targeting control (CRISPRi FC Ci/NT; red). Dominant enhancers based on contact frequency are represented with a red arrow. **B** Correlation of CRISPRi-induced gene-knockdown and chromatin loop dynamic (standard deviation in time) contact frequency for each ARBS. The solid line represents the mean, and the error bars indicate the 95% confidence. The significance was assessed by Spearman's rank correlation ($p_{H3K27ac}$ -0.0003, $p_{H3K4me3}$ -0.002) (ns $p > 0.05$, *$p < 0.05$, **$p < 0.01$, ***$p < 0.001$, ****$p < 0.0001$).

and ARBS2) (Fig. 5E). Supporting the dominance model, we observed an inverse correlation between the inhibition of androgen-induced gene expression and the dynamic change in contact frequency of loops during androgen stimulation (Fig. 6B). This correlation was greater than any other genomic features including AR, FOXA1, H3K4me3, H3K27ac, and chromatin accessibility (Supplementary Fig. 12). These results demonstrate that not all CREs contribute equally to androgen stimulation and that their contact frequency with promoters correlates with their impact on gene expression. Overall, these results underscore the importance of spatial genome organization in transcriptional regulation and validate our proposed multi-enhancer dominance model.

## Discussion

Transcription factors bind to specific DNA sites and regulate gene expression through the recruitment of co-regulatory proteins that activate transcriptional machinery[45,46]. Yet despite extensive research, many questions remain about how this complex process occurs, particularly as there are multiple CREs that interact with each promoter. Using the ligand-activated AR as a model system, we characterized how transcription factor activation changes epigenetics and chromatin looping. Similar to published studies, our work showed that the AR stabilizes/recruits FOXA1 and increases both H3K27ac and chromatin accessibility (Fig. 2A)[2,5,7,47,48]. By characterizing a kinetic dataset, we showed that the change in chromatin accessibility occurs after both FOXA1/AR binding and H3K27ac post-translational modifications, suggesting that this is mediated by the recruitment of additional chromatin-remodeling proteins. Those enhancers that are not bound by AR do not exhibit these changes. However, these epigenetic changes do not solely drive gene expression, as we observed a consistent pattern in all AR-bound CREs, regardless of either the direction of expression (upregulated and downregulated), or timing of maximal nascent transcription (Figs. 2, 4B). Particularly, the regulation of enhancer-associated H3K27ac by AR is independent of gene expression, similar to a recent study[49]. On the other hand, the slight decrease in promoters does not appear to be directly mediated by AR binding, as it was also observed in AR-independent promoters. This suggests a potential competition for histone acetyltransferases[50], especially since we noted an increase in the signal at AR-bound enhancers (Supplementary Fig. 13). However, AR binding selectively induced nascent eRNA transcription at enhancers looped to upregulated genes and not downregulated genes (Fig. 2A). This suggests a regulatory role of eRNA on gene expression and further indicates AR regulates target genes through these looped enhancers. These observations suggest a sequential mechanism, independent of gene expression, in which AR activation recruits specific co-regulatory proteins that alter histone modification and chromatin accessibility.

This work also characterized how AR binding affects genome organization. This is a controversial field, with earlier studies proposing that steroid hormone receptors significantly reorganize chromatin looping when activated[51]. However, recent work has suggested that gene expression may occur through already-existing interactions in both GR-mediated[23] and ER-mediated[52] transcription. Our research found that androgen treatment does not cause the rewiring of chromatin contact loops but instead increases the contact frequencies of previously established loops (Fig. 3C, D and Supplementary Fig. 6). Interestingly, we observed that maximal chromatin looping happens either during or after the maximal nascent gene expression (Fig. 4C). This suggests that increasing chromatin looping does not precede, but instead likely occurs simultaneously with gene transcription. Supporting this result, recent work observed that higher nascent RNA production is associated with a higher frequency of chromatin contacts[22]. Changes in the chromatin contact frequency have also been shown to be associated with differential gene expression[20,23]. Although several studies observed temporal changes in chromatin looping that occur before maximal RNA expression[21,23], this distinction is likely due to the experimental methodology used, as RNA-seq primarily quantifies mature mRNA, whereas Start-seq captures only nascent mRNA. Highlighting the consistent pattern of epigenetic features of AR-bound enhancers (Figs. 2, 4B), we can infer chromatin looping is an additional mechanism which regulates gene expression independently of AR binding. The importance of chromatin loop contact frequency is highlighted when we look at AR-downregulated genes, which show similar changes in epigenetic alterations and chromatin accessibility but not contact frequency (Figs. 2B, 3E). This suggests that AR binding recruits additional factors that potentially increase the contact frequencies of pre-established loops between enhancers and their target gene promoters to regulate the gene expression.

Numerous studies demonstrate that multiple enhancers contribute to the expression of a single gene[7,53]. However, there is no consensus about how each CRE contributes to gene expression. Our work suggests that connected enhancers contribute unevenly, and

there exist dominant loops that have the largest impact on gene expression. These findings align with the assumptions made in the activity-by-contact (ABC) model[54], which presumes that the enhancer's impact on gene expression is correlated with the strength of the contact between them. Both our validation (Fig. 6), and a recent large-scale CRISPRi study[22], demonstrate that those enhancers with higher contact frequency to target promoters are more likely to be functionally important (Fig. 5D). Given multi-enhancer hubs could be a strategy for phenotypic robustness[55], cancer cells could exploit this model to achieve heterogeneous regulation across different cellular contexts. For example, in castration-resistant prostate cancer, various combinations of development enhancers could be activated[26]. While speculative, cells might also hijack these "dominant" enhancers within extrachromosomal DNA to further modulate gene expression[56]. However, further improvements of this model can help us to understand the effect of individual CREs, thus allowing us to evaluate how multiple transcription factors impact gene expression.

Given the complex nature of this system, we had to make several assumptions. First, we opted to investigate AR-mediated transcription using the LNCaP prostate cancer cell line, a well-established androgen-dependent PCa model[57]. While the LNCaP cell line harbors a T878A ligand-binding domain mutation in the AR gene, this mutation does not significantly impact AR function[58]. Next, we chose to define individual regulatory units based on chromatin accessibility, since defining enhancers and promoters by histone modifications is prone to false positives[59,60]. Within these CREs, we focused on hallmark androgen response genes (Fig. 1E), as these have been shown to be regulated by AR. Further, we used HiChIP, a protein-centric HiC method, rather than conventional HiC to provide enhanced resolution and specificity in capturing protein-bound chromatin[61]. By characterizing both promoter-centric (H3K4me3) and enhancer-centric (H3K27ac) chromatin loops, we believe this provided us with different genomic perspectives of chromatin loops and reduced potential biases (Fig. 1B, C)[21,62,63]. We showed that changes in the ChIP-seq signal cause minimal bias on HiChIP contact frequency change (Supplementary Fig. 14). It is important to note that while H3K27ac is strongly impacted by AR binding, we observed that H3K4me3 was relatively static (Figs. 2, 4B), suggesting that the change in contact frequency (Figs. 3C–E, 4C) following androgen treatment was not due to capture efficiency. Moreover, as gene expression regulation circuits are challenging to integrate due to the intricate network structures formed by diverse CREs (Fig. 1D), we utilized regulatory networks wherein the nodes and edges represent the CREs and chromatin loops, respectively. Several studies have employed graph-based approaches to connect genomic regions and address various high-dimensional biological inquiries[64–66]. The versatility of this approach underscores the quantitative advantages of utilizing graphs to characterize chromatin loops. Finally, we investigated a multi-enhancer model by analyzing the correlation between contact frequency and gene expression across all genes at all time points (Fig. 5B and Supplementary Fig. 9). This methodology enabled us to examine the multi-enhancer model AR-agnostic, thereby enhancing the generalizability of our model. This approach also allowed a more in-depth exploration of how AR operates within a gene's context. Despite these limitations, this study represents one of the largest experimental datasets (Fig. 1A) to characterize AR-mediated transcription.

Overall, this paper provides insight into several important aspects of AR-mediated gene expression. We show that AR binding triggers a temporal cascade which increases FOXA1 and H3K27ac that affects chromatin accessibility. Further, we demonstrate that AR does not introduce novel chromatin loops, but instead increases the contact frequency between AR-bound enhancers and their target promoters. However, the effect of each enhancer on gene expression is markedly heterogeneous and proportional to promoter contact frequency. These findings suggest that while AR binding to DNA induces a stepwise epigenetic alteration, the impact of bound enhancers is strongly dependent on the contact frequency of the established chromatin loops with the target promoter.

## Methods

### Experimental

**LNCaP cell culture and DHT treatment.** LNCaP cells (#CRL-1740, ATCC) were grown in phenol red-free RPMI (#11835030, GIBCO) with 10% charcoal-stripped FBS (#100–119, GemBio) for 3 days and then were stimulated with 10 nM dihydrotestosterone (DHT) (A8380, Sigma) for 0.5, 4, 16, or 72 h. For the vehicle samples, the cells were treated with 100% EtOH. Subsequently, cells were collected for further experiments (ChIP-seq, RNA-seq, ATAC-seq, HiChIP, or Start-seq) accordingly[67]. LNCaP cells were authenticated by sequencing and comparing short tandem repeats to parental LNCaP cells in the ATCC database. Prior to experiments, cells were tested for mycoplasma contamination with LookOut Mycoplasma PCR Detection Kit (Sigma-Aldrich #D9307).

**Chromatin immunoprecipitation assays with sequencing (ChIP-seq).** ChIP-seq in LNCaP was performed as previously described in ref. 67. Briefly, 10 million cells were fixed with 1% formaldehyde at room temperature for 10 min, quenched, and collected in lysis buffer (1% NP-40, 0.5% sodium deoxycholate, 0.1% SDS and protease inhibitor [#11873580001, Roche] in PBS). Chromatin was sonicated with a Covaris E220 sonicator (140 PIP, 5% duty cycle, 200 cycle burst). The sample was then incubated with antibodies (AR: Abcam ab133273, FOXA1: Abcam ab23738, H3K27ac: Diagenode C15410196; H3K4me3: Diagenode C15410003) coupled with Dynabeads protein A and protein G beads (Life Technologies) at 4 °C. Incubated chromatin was washed with RIPA wash buffer (100 mM Tris pH 7.5, 500 mM LiCl, 1% NP-40, 1% sodium deoxycholate) for 10 min six times and rinsed with TE buffer (pH 8.0) once. DNA was purified using a MinElute column followed by incubation in the de-crosslinking buffer (1% SDS, 0.1 M NaHCO3 with Proteinase K and RNase A) at 65 °C. Eluted DNA was prepared as sequencing libraries with the ThruPLEX-FD Prep Kit (Takara bio, # R400675). Libraries were sequenced with 150-BP PE on an Illumina HiSeq 2500 Sequencing platform at Novogene.

**RNA-seq.** LNCaP cells ($5 \times 10^5$) were harvested for RNA-seq[68]. Total RNA was collected from the cells using an RNeasy kit (Qiagen, #74104,) with DNase I treatment (Qiagen, #79254). The library preparations, quality control, and sequencing on HiSeq 2500 Sequencing platforms (150-BP PE) were performed by Novogene.

**Assay for transposase-accessible chromatin with sequencing (ATAC-seq).** LNCaP cells were isolated and subjected to modified ATAC-seq as previously described in ref. 67. Briefly, 50,000 nuclei were pelleted and resuspended in ice-cold 50 μl of lysis buffer (10 mM Tris-HCl, pH 7.5, 10 mM NaCl, 3 mM MgCl2, 0.1% NP-40, 0.1% Tween20, and 0.01% Digitonin). The subsequent centrifugation was performed at 500×g for 10 min at 4 °C. The nuclei pellets were resuspended in 50 μl of transposition buffer (25 μl of 2× TD buffer, 22.5 μl of distilled water, 2.5 μl of Illumina Tn5 transposase) and incubated at 37 °C for 30 min with shaking at 1000 rpm for fragmentation. Transposed DNA was purified with the MinElute PCR Purification kit (Qiagen). DNA was purified using Qiagen MinElute (#28004), and the library was amplified up to the cycle number determined by 1/3rd maximal qPCR fluorescence. ATAC-seq libraries were sequenced with 150-BP PE high-throughput sequencing on an Illumina HiSeq 2500 Sequencing platform (Novogene).

**HiC combined with capture ChIP-seq (HiChIP).** HiChIP was performed as previously described in ref. 68. Trypsinized LNCaP cells (10 million) were fixed with 1% formaldehyde at room temperature for

10 min and quenched. The sample was lysed in HiChIP lysis buffer and digested with MboI (NEB) for 4 h. After 1 h of biotin incorporation with biotin dATP, the sample was ligated with T4 DNA ligase for 4 h with rotation. Chromatin was sonicated using Covaris E220 (conditions: 140 PIP, 5% DF, 200 CB) to 300–800 bp in ChIP lysis buffer (1% NP-40, 0.5% sodium deoxycholate, 0.1% SDS and protease inhibitor in PBS) and centrifuged at 13,000 rpm. for 10 min at 4 °C. Preclearing 30 μl of Dynabeads protein A/G for 1 h at 4 °C was followed by incubation with antibodies (H3K27ac, Diagenode, C15410196; H3K4me3, Diagenode C15410003). Reverse-crossed IP sample were pulled down with strep-tavidin C1 beads (Life Technologies), treated with Transposase (Illumina) and amplified with reasonable cycle numbers based on the qPCR with a five-cycle pre-amplified library. The library was sequenced with 150-BP PE reads on the Illumina HiSeq 2500 Sequencing platform (Novogene).

**Small capped nascent RNA sequencing (Start-seq).** For Start-seq, LNCaP cells were grown and collected as described above. Cell pellets were washed with ice-cold 1x PBS. One million cells were treated with 1.5 mL of Nun Buffer (0.3 M NaCl, 1 M Urea, 1% NP-40, 20 mM HEPES pH 7.5, 7.5 mM $MgCl_2$, 0.2 M EDTAm, protease inhibitors, and 20 U/mL SUPERase-IN) for 30 min on ice with frequent vortexing. Chromatin was pelleted by centrifugation at $12,500 \times g$ for 30 min for 4 °C. After three times-washing with 1 mL ice-cold chromatin washing buffer (50 mM Tris-HCL pH 7.5 and 40 U/mL SUPERase-In) and additional 0.5 mL of Nun buffer. After centrifugation for 5 min at $500 \times g$, 0.5 m TRIzol was added to the remaining pellet. Libraries were prepared according to the TruSeq Small RNA Kit (Illumina). To normalize samples, 15 synthetic capped RNAs were spiked into the Trizolpreparation at a specific quantity per $10^7$ cells, as in ref. [69]. The library was sequenced with 150-BP PE reads on the Illumina HiSeq 2500 Sequencing platform (Novogene).

**CRISPRi.** Guide RNAs (gRNAs) were designed for each enhancer and promoter region using CRISPR-SURF[70]. Cas-OFFinder was used to eliminate off-target gRNAs[71]. LNCaP cells stably expressing dCas9-KRAB (Addgene #89567) were seeded in a six-well plate at a density of 200 K cells per well. For transfection, a total of 500–1500 ng DNA was used, divided according to the number of available gRNAs. Transfection was performed using Mirus TransIT-X2. After transfection, the media was replaced, and 2 ng/μl of puromycin was added for selection. Following 72 h of selection, the medium was changed to charcoal-stripped serum for androgen deprivation. After 48 h, the cells were treated with 10 nM DHT for 4 h and then trypsinized. RNA extraction and cDNA preparation was performed using the LunaScript® RT SuperMix Kit. The androgen-induced expression was quantified using qRT-PCR, and each experiment was conducted in triplicate. The sequences of gRNAs and qRT-PCR primers can be found in Supplementary Data 1.

## Bioinformatics analyses
**RNA-seq analysis.** Reads were aligned to the hg19 human genome with STAR (v2.7.9)[72] with quant mode (--quantMode TranscriptomeSAM). Next, "toTranscriptome" bam files were fed into Salmon (v0.14.1)[73] to quantify TPM values for (Gencode v19)[74] transcripts. To generate the signal track files from RNA-seq, VIPER[75] is used. Briefly, STAR, as the default aligner, converts BAM files into BigWig files using Bedtools (v2.30.0)[76].

**ChIP-seq and ATAC-seq analyses.** ChIP-seq and ATAC-seq were processed through the ChiLin pipeline[77]. Briefly, Illumina Casava1.7 software used for base calling and raw sequence quality and GC content were checked using FastQC (v0.10.1). The Burrows–Wheeler Aligner (BWA[78], v0.7.10) was used to align the reads to the human genome hg19. Then, MACS2[79] (v2.1.0.20140616) was used for peak calling with an FDR

q value threshold of 0.01. Bed files and Bigwig files were generated using bedGraphToBigWig[80], and the union of narrow and broad peaks from ChIP-seq were used as anchors to call loops. The following quality metrics were assessed for each sample: (i) percentage of uniquely mapped reads, (ii) PCR bottleneck coefficient to identify potential over-amplification by PCR, (iii) FRiP (fraction of non-mitochondrial reads in peak regions), (iv) peak number, (v) number of peaks with 10-fold and 20-fold enrichment over the background, (vi) fragment size, (vii) the percentage of the merged peaks with promoter, enhancer, intron, or intergenic, and (viii) peak overlap with DNase I hypersensitivity sites. For datasets with replicates, the replicate consistency was checked by two metrics: (1) Pearson correlation of reads across the genome using UCSC software wigCorrelate after normalizing signal to reads per million and (2) percentage of overlapping peaks in the replicates.

**HiChIP loop calling.** After trimming adapters from the HiChIP datasets using Trim Galore (v0.5.0) (https://github.com/FelixKrueger/TrimGalore), we used HiC-Pro (v3.1.0)[81] as previously described in ref. [68]. This aligned the reads to the hg19 human genome, assigned reads to MboI restriction fragments, and removed duplicate reads. We used the following options: <MIN_MAPQ = 20, BOWTIE2_GLOBAL_OPTIONS = −very-sensitive−end-to-end−reorder, BOWTIE2_LOCAL_OPTIONS = −very-sensitive−end-to-end−reorder, GENOME_FRAGMENT = MboI_resfrag_hg19.bed, LIGATION_SITE = GATCGATC, LIGATION_SITE = "GATCGATC," BIN_SIZE = "5000."> All other default settings were used. To build the contact maps, the HiC-Pro pipeline selects only uniquely mapped valid read pairs involving two different restriction fragments. We applied FitHiChIP (v10.0)[82] for bias-corrected peak and DNA loop calls. FitHiChIP models the genomic distance effect with a spline fit, normalizes for coverage differences with regression, and computes statistical significance estimates for each pair of loci. We used the FitHiChIP loop significance model to determine whether interactions are significantly stronger than the random background interaction frequency. As anchors to call loops in the HiChIP analyses, we used 842,367 regions for H3K27ac and 136,939 regions for H3K4me3, which resulted from merging ChIP-seq narrow and broad peaks comprised two replicates for each broad and narrow peak and each of the five-time points (0 m, 30 m, 4 h, 16 h, 72 h) for H3K27ac, and one replicate for each broad and narrow peak and each of the same time points for H3K4me3. We used a 5 kb resolution and considered only interactions between 5 kb and 3 Mb. We used the peak to all for the foreground, meaning at least one anchor needed to be in the peak rather than both. The corresponding FitHiChIP options specification is <IntType=3> For the global background estimation of expected counts (and contact probabilities for each genomic distance), FitHiChIP can use either peak-to-peak (stringent) or peak-to-all (loose) loops for learning the background and spline fitting. We specified the suggested option to merge interactions close to each other to represent a single interaction when their originating bins are closer. The corresponding FitHiChIP options specifications are <UseP2PBackgrnd = 0> and <MergeInt = 1> (FitHiChIP (L + M)). We used the default FitHiChIP q value <0.01 to identify significant loops. We used hicpro2higlass (v3.1.0) to convert allValidPairs to.cool files after having specified the hg19 chromosome sizes, using the following command: <hicpro2higlass.sh -i sample.allValidPairs -r 5000 -c chrsizes -n> The reads from HiChIP data were merged from every time point for H3K27ac and H3K4me3 separately, and the reference loop sets were called with the same parameters above (resulting loop number, n = 296,326; n = 278,491, respectively). Next, the contact frequency values of each time point at loops are captured from the.cool files using the Python package, Cooler (v0.9.3). The count table was then TMM normalized among time points to reduce the between-sample variation.

**Annotation of cis-regulatory elements/defining background genes.** CREs were defined as ±2.5 kb around the summit of the accessible

region from ATAC-seq peaks at any given time point. Promoters were defined with a multi-step process. First, we identified the highest expression transcript (Gencode v19)[74] isoforms using Salmon (v0.14.1)[73], see RNA-seq analysis. Next, the start locations -according to strand information- of the highest expressing transcripts were collected, and the summit was extended +/2.5 kb. If they overlap with a defined CRE, we assign the overlapping CRE as the active promoters. From these annotated active promoters, the genes of HALLMARKS_ANDROGEN_RESPONSE from the Molecular Signature Database (MSigDB)[83] were defined as AR-regulated genes. The transcriptome was divided into four quartiles, and AR-regulated genes were compared with similarly expressing genes (Supp Fig. 1A, B). Overlapping CREs to AR ChIP-seq peaks at any time point were defined as potential AR-bound enhancers. The median number of AR-bound enhancers (E + AR; ~2) of AR-regulated genes were less than AR-free enhancers (E-AR; ~14) (Supp Fig. 2C). Similarly, a CRE was considered FOXA1-bound, if it intersects with a FOXA1 peak at any time point (Supp Fig. 2A). Based on the contact frequency change results from the AR-upregulated genes, we defined the downregulated genes at the 16 h time point, as it exhibited the most marked regulatory response. To identify downregulated genes, we calculated the log2foldChange induction between 16 h and 0 m, considering those with a value below −1 as downregulated genes (Supp Fig. 4).

**Constructing the graph network.** When we called significant loops from each time point separately, there were few called loops overlapping between time points, potentially due to both the loop calling methodologies and experimental noise present in HiChIP data[84]. As we could observe the matching loops in the contact matrices of all time points (Supp Fig. 6), we instead generated a reference loop set, normalized the count matrix, and compared contact frequencies similar to published work[20]. A custom R (v4.1.1) script (https://github.com/lacklab/AR_transcription) was used to extract interaction pairs of annotated CRE regions (BED) according to the reference loop set (BEDPE; both H3K27ac and H3K4me3) using "GenomicInteractions" (v1.34)[85]. The graph structure is built in custom Python (v3.9.7) script (link) using the NetworkX (v3.1) package[86]. Briefly, any pair of CREs are included as nodes in the network with TMM normalized HiChIP contact frequency as weighted edges, for both H3K27ac and H3K4me3 at every time point.

**Epigenetic changes of CREs and kinetic changes of enhancers of AR-regulated genes.** The average signal (AR, FOXA1, H3K27ac, H3K4me3, ATAC) at the promoters and CREs were collected using a custom Python package [https://github.com/birkiy/bluegill.git]. To reduce the batch effect across time, TMM normalization was applied individually for each epigenetic factor[87]. To visualize the temporal change upon androgen stimulation, line plots were drawn (Fig. 2). For the selected six AR-regulated genes (Fig. 3F), both the average signal (AR, FOXA1, H3K27ac, H3K4me3, and ATAC) over the first-degree interacting enhancers of each gene promoter and contact frequency (CF) between every gene promoter and corresponding enhancer were collected for every time point. The standard deviation (SD) of each feature along the time domain was calculated, and the min-max was normalized.

**Calculation of chromatin contact frequency change.** Promoter-centric analysis was done with the query loop sets, AR-regulated genes' promoter loops (P + AR), and random highly expressed genes' promoter loops (P-AR; $n = 100$; seed = 7). This was compared to reference loop sets ($n = 100$) that were randomized (1000 iterations; seed = 7) from highly expressed genes' promoter loops (Fig. 3B). Fold change was calculated for each iteration between the average contact frequency of loops in the query and reference. The same approach was

used to calculate the contact frequency fold change from the enhancer viewpoint. The loop sets were selected according to the E-P pairs. While selecting the query and reference loops, the number of randomly selected promoters was fixed ($n = 100$). The loops between AR-regulated gene promoters to AR-bound (E + AR) and AR-free enhancers (E-AR), and between AR-independent gene promoters to AR-free enhancers (E-AR) were compared to randomized loops from an AR-free enhancer to highly expressed gene promoters (1000 iterations; seed = 7). Due to the limited sample sizes and variability in AR-upregulated genes across different quartiles, we chose to analyze these together to increase the overall trend. However, we also provided the fold changes in contact frequency for each quartile of AR-upregulated genes (Supp Fig. 7). This analysis confirms that contact frequency generally increases across quartiles, even with a diminishing effect.

**Start-seq analysis.** Adapter sequences and low-quality 3′ ends were removed from paired-end reads of all samples using cutadapt (v1.2.1). Reads shorter than 20 nucleotides were discarded (-m 20 -q 10), and a single nucleotide was trimmed from the 3′ end of all remaining reads to enable successful alignment with bowtie (v1.1.1). The first in pair flagged reads were filtered to generate signal (bigwig) files for each strand using bedtools genomecov (-bg -5 -strand ±, respectively).

The transcripts (Gencode v19) were extracted GTF file. The maximum signal within a range of ±500 bp of TSS was gathered from the forward Start-seq track for plus-stranded transcripts, and the reverse for minus-stranded transcripts. Next, the log2foldchange (LFC), compared to 0 m, was calculated for every time point (30 m, 4 h, 16 h, 72 h). If a transcript was found to be LFC >1 at any time point, it was considered as differential upregulation. Next, the nascent expression levels at all time points for those transcripts (the union of differential upregulation) were z-normalized to capture the highest expression time point, which is assigned to time-based expression groups (Fig. 4A). Similar to enhancer viewpoint analysis, the first-degree AR-bound enhancer contact frequency to the gene promoters was compared to randomly selected contact frequencies (1000 iterations) between the highly expressed gene promoters ($n = 100$) and their first-degree AR-free enhancers.

**Contact frequency, expression correlation.** To explore the contact frequency vs. expression, we performed three first-degree summarization models (average, maximum, and sum). Briefly, we identified all first-degree contacts of every gene promoter, then we summarized the contact frequencies with the corresponding function of the models for every gene. To reduce the signal noise, we first sorted genes according to expression, binned them into equal-sized sets ($k = 25$), plotted bins as a scatter plot with average log expression on the x-axis, and averaged summarized contact frequency on the y-axis (Fig. 5B and Suppl Fig. 9).

**Random forest regressor.** The binned ($k = 25$) contact frequency and expression data (see above) were used to train a random forest regressor from the Sklearn (v1.3.1)[88] with "random_state=0". The permutation importance of each feature is also calculated within the Sklearn[43].

**Gini index analysis.** The chromatin contact frequency between gene promoter and first-degree interacting elements was identified and the Gini index was calculated for a 16 h time point (Fig. 5C). To calculate the Gini index a custom python function was utilized [https://github.com/birkiy/cisregulatorynetworks.git]. The mean absolute difference is the average absolute difference between all pairs of items in the population. The relative mean absolute difference is obtained by dividing the mean absolute difference by the population's average ($x$)

to account for differences in scale (e.q. 1), where $x_i$ is the contact frequency of loop $i$, and there are $n$ loops of a promoter.

$$G = \frac{\sum_{i=1}^{n}\sum_{j=1}^{n}|x_i - x_j|}{2n^2\underline{x}} \qquad (1)$$

**Circle plots.** Circle plots were drawn with NetworkX and Matplotlib libraries[86,89] [https://github.com/birkiy/cisregulatorynetworks.git].

## Statistical analysis

The probability of finding $k$ loops from each CRE was determined by hypergeometric distribution (Supp Fig. 5). When comparing the target set with the background set, a non-parametric Mann–Whitney $U$-test was applied. Correlation coefficients and their $p$ values were calculated with Spearman's test. All statistical tests were performed using the SciPy (v1.11.1) Python package[90].

## Reporting summary

Further information on research design is available in the Nature Portfolio Reporting Summary linked to this article.

## Data availability

The generated datasets are deposited to Gene Expression Omnibus (GEO), and are publicly availiable through GSE251898 accession (ATAC-seq: GSE251893, ChIP-seq: GSE251894, HiChIP: GSE251895, RNA-seq: GSE251896, Start-seq: GSE251897). Additional data were provided with this paper through Zenodo [https://doi.org/10.5281/zenodo.13770484]. Source data are provided with this paper.

## Code availability

The code to reproduce the results from this study's data were available through the project's source code repository [https://github.com/birkiy/cisregulatorynetworks.git] and [https://github.com/birkiy/bluegill.git].

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

## Acknowledgements

N.A.L. was supported by funding from TUBITAK (221Z116), W81XWH-21-1-0234 (DoD), and CIHR PJT-173331. M.L.F. was supported by the Claudia Adams Barr Program for Innovative Cancer Research, the Dana-Farber Cancer Institute Presidential Initiatives Fund, the H.L. Snyder Medical Research Foundation, the Cutler Family Fund for Prevention and Early Detection, the Donahue Family Fund, W81XWH-21-1-0339, W81XWH-22-1-0951 (DoD), NIH Awards R01CA251555, R01CA227237, R01CA262577, R01CA259058, and a Movember PCF Challenge Award.

## Author contributions

Conceptualization, M.L.F. and N.A.L.; Methodology, U.B.A., J.S., and C.G.; Validation, D.O., Formal analysis, U.B.A., C.G., and G.M.N.; Investigation, J.S., D.O., and S.R.G.; Resources, K.A.; Data curation, J.S., B.J.F., and U.B.A.; Writing—original draft, U.B.A.; Writing—review and editing, J.S., C.G., D.O., B.J.F., G.M.N., S.R.G., K.A., F.H., M.F.L., and N.A.L.; Visualization, U.B.A. and N.A.L.; Supervision, K.A., F.H., M.F.L., and N.A.L.; Funding acquisition M.F.L. and N.A.L.

## Competing interests

The authors declare no competing interests.

## Additional information

[1]Vancouver Prostate Centre, University of British Columbia, Vancouver, BC V6H 3Z6, Canada. [2]Center for Functional Cancer Epigenetics, Dana-Farber Cancer Institute, Boston, MA 02215, USA. [3]Integrative Data Analysis Unit, Health Data Science Centre, Human Technopole, Milan 20157, Italy. [4]Department of Pathology and Laboratory Medicine, David Geffen School of Medicine, University of California, Los Angeles, Los Angeles, CA 90024, USA. [5]Department of Biological Chemistry and Molecular Pharmacology, Harvard Medical School, Boston, MA 02115, USA. [6]Department of Biomedical Informatics, Harvard Medical School, Boston, MA 02115, USA. [7]The Eli and Edythe L. Broad Institute, Boston, MA 02142, USA. [8]Department of Urologic Sciences, University of British Columbia, Vancouver, BC V5Z 1M9, Canada. [9]Department of Computer Science, University of British Columbia, Vancouver, BC V6T 1Z4, Canada. [10]Department of Medical Oncology, Dana-Farber Cancer Institute, Boston, MA 02215, USA. [11]Department of Medical Pharmacology, School of Medicine, Koç University, Istanbul 34450, Turkey. [12]Koç University Research Centre for Translational Medicine (KUTTAM), Koç University, 34450 Istanbul, Turkey. [13]These authors contributed equally: Umut Berkay Altıntaş, Ji-Heui Seo. [14]These authors jointly supervised this work: Matthew L. Freedman, Nathan A. Lack. ✉e-mail: nlack@prostatecentre.com

