## [Peer Review file · Nature Communications]

Decoding the Epigenetics and Chromatin Loop Dynamics of Androgen Receptor-Mediated Transcription

Corresponding Author: Dr Nathan Lack

Version 0:

Reviewer comments:

Reviewer #1

(Remarks to the Author)

This manuscript is the first to define the relationship between chromatin loop dynamics and the AR-regulated gene expression. It builds on a principle previously established by Greenwald et al., in a paper in this same journal in 2019. By applying it to the important question of how the AR regulates gene expression in a prostate cancer cell-line the authors are furthering our understanding of AR activity. As a centrally important transcription factor for prostate cancer development this is of wide interest and relevance. The data are clearly presented and the conclusions are supported by the analyses undertaken. Genetic variants can also influence chromatin loop dynamics. It would therefore be helpful for the authors to highlight loci within their datasets that harbor prostate cancer risk variants. This will make it easier for researchers working on the functional contribution of these variants to prostate cancer development to select candidates for further study. This could be included in the supplementary information.

Reviewer #2

(Remarks to the Author)

An interesting and thorough paper throwing more light on the complexities of hormonal control of transcription - here for the androgen receptor but likely to have wider implications.

Some clarifications and adjustments needed:

- Page 6: Annotation of Cis-Regulatory Elements/Defining Background genes
Needs some clarity on how AR up- and down-regulated genes were defined in this study.
From what I understand, AR upregulated genes were defined as those from the HALLMARKS_ANDROGEN_RESPONSE geneset that were associated with an active promoter (as identified in this study). There are 101 genes in the Hallmarks Androgen response geneset and 89 genes denoted as AR regulated in Supp fig 1A. Were only 89 genes from the HALLMARKS geneset associated with an active promoter? Were the remaining 12 genes excluded because they were not associated with an active promoter or because they were down-regulated? What were the timepoints and log fold change cutoffs for considering up-regulated genes?
For downregulated genes, the authors state "To identify down-regulated genes, we calculated the log2foldChange induction between 16h and 0m, considering those with a value below -1 as downregulated genes". Are all of these genes considered AR downregulated, although they might not be directly regulated by the AR? And why the 16h timepoint?
- Supplementary Figure 1 A,B: Clarify if these are AR-upregulated
- Supplementary Figure 1D: Needs labels above graphs and on axes
- Supplementary Figure 2: Track labels not legible. Also, purple and black tracks not defined in figure legend.
- Page 9, Lines 436-437:
Does contact frequency increase similarly/equally in all four quartiles of upregulated genes?
- Page 10, lines 449-450:

Is there any potential explanation for the subset of AR bound enhancers (E+AR) that had the highest change in epigenetic features and contact frequency? E.g presence of a motif that may suggest they are all regulated by the same cofactor?

• Page 10, Association of nascent transcription to epigenetic changes and contact frequency:
Were the authors able to detect enhancer RNA expression with their start-seq dataset, and does eRNA expression correlate contact frequency or dominant loops?

• Line 499-500: CRE-promoter interactions of androgen-regulated genes (n=88) but its 89 in Supp Fig 1 A,B. Also, I couldn't find a Supp Table 2. Is it labelled as Supp Table 1a in the attached files?

Reviewer #3

(Remarks to the Author)

Title: Decoding the Epigenetics and Chromatin Loop Dynamics of Androgen Receptor-Mediated Transcription
Manuscript Summary:

In this manuscript, Umut et al, have assessed the impact of AR bound CRE on AR regulated gene expression in prostate cancer cells. The authors utilized a multiomic approach to investigate AR induced epigenetic and chromatin loop changes that can affect gene expression. The following key findings were reported in the manuscript.

- 1.AR induces sequential epigenetic changes in CREs, which does not affect gene expression
- 2.AR binding does not affect pre-existing chromatin loops but increases its contact frequency with target promoters
- 3.Proposed a unbalanced/heterogeneous multi-enhancer dominant model

While the findings presented in the study offer mechanistic insights into AR's transcriptional regulation, thereby adding to our understanding of AR's function, they are not novel but rather build upon pre-established models (PMID: 31294690; <https://doi.org/10.1186/s13059-021-02322-1>). Hence, it is challenging to ascertain the significance of the findings presented in the study. Furthermore, several aspects in the manuscript require clarification, and the authors should include additional information to underscore the significance of their findings."

Attached are the positive reviews and some of the major limitations observed in the study throughout the manuscript.

Positive reviews:

- 1.The manuscript is written in a concise fashion and the findings are presented clearly
- 2.The conclusions that the authors make is supported by the results presented in the manuscript
- 3.The methodology used in the manuscript are straightforward. The techniques employed in the manuscript are familiar and widely accepted within the field of omics research. This suggests that the study's methods are robust and likely to be reproducible, enhancing the credibility of the findings.

Major Limitations:

- 1.It is crucial for the authors to address the gap in knowledge regarding whether the proposed multi-enhancer model can account for transcriptional regulation of AR independent of cell types. Given that the authors have solely utilized LNCaP cells for deriving the bulk sequencing data, it prompts the question of whether this finding is unique to cancer cells or if it extends to other cell types including non-cancerous prostate cells."
- 2.What accounts for the differential H3K27ac levels in AR-bound enhancer regions compared to AR-regulated promoter regions (Figure 2) and its potential impact on AR-regulated gene transcription? Some studies suggest that H3K27ac does not significantly affect the transcriptome (PMID: 32085783), while others indicate crosstalk between H3K27ac and H3K4me3 (<https://doi.org/10.1038/s41598-021-95398-5>). I hope the authors can further contribute to this discussion.
- 3.No validations were conducted to confirm any of the AR regulated gene expression (downregulated and upregulated) shown in the bulk data throughout the study.
- 4.All the bulk datasets used in the study are derived from the prostate cancer cell line LNCaP. It is important to note that AR in LNCaP harbors a missense mutation (AR-T857A), resulting in broad and promiscuous steroid and anti-steroid binding specificity (PMID: 1562539 and PMID: 15657128). Will the findings in this study change accounting for this mutation and by the fact that AR in LNCaP can be activated by other ligands other than testosterone or DHT.
- 5.There are known androgen receptor variants (AR-V) with truncated ligand-binding domain that are constitutively active. The findings presented in this manuscript correspond to steroid induced AR (assuming androgens induce AR homodimerization) transcriptional regulations. How will the model presented in the study account for AR-V regulated transcription specifically in castration condition.
- 6.Specifically in castration condition, we observe copy number gains in AR gene (which can result in the formation of AR gene containing extrachromosomal DNA) and copy number gains in enhancer regions (<https://doi.org/10.1038/s41467-023-40315-9>; PMID: 29909987). How will one apply the multi-enhancer model in this case scenario.
- 7.In the article 'Multi-enhancer transcriptional hubs confer phenotypic robustness,' the authors suggest the existence of multi-enhancer hubs that facilitate phenotypic robustness under stress. I would like to raise the question to the authors: Does the multi-enhancer dominant model facilitate phenotypic robustness, and does it have an impact on the development of castration resistance? Addressing this question could be pivotal in demonstrating the translational significance of the findings in this paper.

Reviewer #4

(Remarks to the Author)

In this manuscript, the authors explored the effect of androgen on chromatin looping and gene expression in LNCaP cells through a multi-omic approach, including ChIP-seq, ATAC-seq, HiChIP, and Start-seq. They showed that AR binding to DREs does not alter chromatin loops but increases the contact frequency of existing loops, suggesting that the effect of AR-bound enhancers on gene expression correlates with their contact frequency with gene promoters. Additionally, by using CRISPR-SURF to target specific enhancers or promoters of AR regulated genes, the authors observed a moderate decrease in gene expression, supporting their model.

Overall, the manuscript is well written, and the results are well organized. However, this study appears to contribute only marginally to the understanding of AR-mediated transcriptional regulation and chromatin looping. In particular, it has several limitations:

The observed increase in looping frequency primarily relies on H3K27ac HiChIP data. However, upon androgen treatment, there's a notable increase in H3K27ac levels at enhancers of AR-activated genes (as shown in Fig.2). This increase could potentially cause bias on chromatin loop calculation. Despite the authors' argument to the contrary, using H3K4me3 as a comparison may not be appropriate since its ChIP-seq signals remain static under androgen treatment. The potential bias could impact the study's major conclusion.

The research's scope is quite limited, focusing solely on a single AR-positive prostate cancer cell line without addressing any physiological relevance of these findings. Even at the cellular level, questions remain about whether these mechanisms are broadly applicable across other steroid receptors or are simply specific to AR.

The AR-regulated, AR-activated, AR-downregulated, and AR-independent genes need be clearly defined and be consistent from figures to figures.

Version 1:

Reviewer comments:

Reviewer #1

(Remarks to the Author)

The authors have addressed the comments satisfactorily.

Reviewer #2

(Remarks to the Author)

All my comments/suggestions have been addressed satisfactorily

Reviewer #3

(Remarks to the Author)

To the Authors,

Thank you for addressing the reviewer comments. I have made my recommendation that the manuscript can be accepted for publication.

The point-by-point response to reviewers:

Reviewer comments are shown in bold italics

Reviewer #1:

Genetic variants can also influence chromatin loop dynamics. It would therefore be helpful for the authors to highlight loci within their datasets that harbor prostate cancer risk variants. This will make it easier for researchers working on the functional contribution of these variants to prostate cancer development to select candidates for further study. This could be included in the supplementary information.

We appreciate the reviewer's suggestion to investigate the loci harboring prostate cancer risk-associated variants. To explore this, we overlaid 451 prostate cancer risk variants (PMID: 37945903) with our kinetic data to identify the looped cis-regulatory elements that contain any of the variants. Our analysis revealed that approximately 24% of these disease-associated variants (108 out of 451) were located in looped CREs. Statistical analysis did not find a significant association between dominance and risk variants (Fisher's exact test $p>0.05$).

CRE	Contains Variant	Not contain a Variant
Dominant	20	8443
Not Dominant	88	41140

Although outside our study's primary focus, we explored specific enhancer-promoter (E-P) interactions with risk variants. We found that the dominant enhancers, which contain the risk variants, interact with 44 gene promoters, including genes like *ETV6*, *NEDD9*, *MEIS1*, and *BCL2L14*, *SMAD6* that have been associated with prostate cancer growth and initiation (PMID: **329455655**, **3632899**, **32681068**, **22546513**, **33174391**). To help guide future studies we have included all prostate cancer risk variants and their dominant loops in a supplementary table (**Supp Table 3**) We hope that these findings will stimulate further research into the mechanisms of these cancer-associated variants.

Reviewer #2:

Page 6: Annotation of Cis-Regulatory Elements/Defining Background genes.

Needs some clarity on how AR up- and down-regulated genes were defined in this study. From what I understand, AR upregulated genes were defined as those from the HALLMARKS_ANDROGEN_RESPONSE geneset that were associated with an active promoter (as identified in this study). There are 101 genes in the Hallmarks Androgen response geneset and 89 genes denoted as AR regulated in Supp fig 1A. Were only 89 genes from the HALLMARKS geneset associated with an active promoter? Were the remaining 12 genes excluded because they were not associated with an active promoter or because they were down-regulated? What were the timepoints and log fold change cutoffs for considering up-regulated genes?

We thank the reviewer for highlighting this crucial point. As the reviewer correctly pointed out, we defined androgen receptor (AR) upregulated genes from the HALLMARKS_ANDROGEN_RESPONSE. We chose to use this well-characterized signature given the potential variability of gene expression and the influence of post-transcriptional regulation that could overcomplicate our analyses (PMID: 26771021, 34496240). While there are 101 HALLMARKS_ANDROGEN_RESPONSE genes, only 89 of the promoters were accessible at any time point in our ATAC-seq data. Of these, 88 exhibited at least one loop to a cis-regulatory element (CRE). Supporting this gene signature, we observed strong upregulation following androgen treatment (Supp Fig 1C, D). We have better described how this gene signature was selected (Page 9; Lines 396-399) and modified the figures accordingly (Supp Fig 1A, B).

For downregulated genes, the authors state “To identify down-regulated genes, we calculated the log2foldChange induction between 16h and 0m, considering those with a value below -1 as downregulated genes”. Are all of these genes considered AR downregulated, although they might not be directly regulated by the AR? And why the 16h timepoint?

This is an excellent point, particularly as the mechanism of AR-dependent gene downregulation is poorly understood (PMID: 17940184, 30606742). We chose to define downregulation at a specific time point as this exhibited the most marked regulatory response based on the contact frequency change results from the AR-upregulated genes. We acknowledge the reviewer’s insight regarding potential indirect AR-mediated regulation and have included additional clarification about both this mechanism and why this time was selected (Page 6; Lines 265-267)

Supplementary Figure 1 A,B: Clarify if these are AR-upregulated

Supp Fig 1A, B emphasized the HALLMARKS_ANDROGEN_RESPONSE genes, which are primarily upregulated (Supp Fig 1C, D). We have made the necessary corrections to the figures.

Supplementary Figure 1D: Needs labels above graphs and on axes

The labels have been corrected (Supp Fig 2C; previously Supp Fig 1D).

Supplementary Figure 2: Track labels not legible. Also, purple and black tracks not defined in figure legend.

Thank you for pointing out this error. The tracks have been labeled and the legend has been corrected (Supp Fig 3; previously Supp Fig 2).

Page 9, Lines 436-437: Does contact frequency increase similarly/equally in all four quartiles of upregulated genes?

We appreciate the reviewer's insightful question. This is something we carefully considered in our analysis. However, due to the limited sample sizes and variability in AR-upregulated genes across different quartiles, we chose to analyze them as a group to observe the overall trend. Our rationale for selecting highly expressed genes as the background gene set was that these genes exhibit the most similar expression patterns and number of loops from promoters as AR-upregulated genes (**Supp Fig 1A, Supp Fig 5B; previously Supp Fig 4B**). To address the reviewer's query, we included a new supplementary figure of the fold changes in contact frequency for each quartile of AR-upregulated genes (**Supp Fig 7**). This analysis confirms that contact frequency generally increases across quartiles, even with a diminishing effect.

Page 10, lines 449-450: Is there any potential explanation for the subset of AR bound enhancers (E+AR) that had the highest change in epigenetic features and contact frequency? E.g presence of a motif that may suggest they are all regulated by the same cofactor?

We thank the reviewer for bringing up this point. The key determinants of these dominant loops are indeed crucial aspects to consider. Although we recognize the potential value of motif analyses in elucidating dominance, our attempts to identify known DNA motif sequences associated with dominant AR-bound enhancers have not yielded promising results. This could be potentially due to the limited number of dominant loops and also the relative size of each called cis-regulatory element. Our resolution is limited to the 5kb window from our HiChIP data, which is approximately 20 times larger than a typical ChIP-seq peak (250bp). This larger window leads to a high number of false positives, as transcription factor motifs are generally 6-12 bp. Even when focusing on the accessible sites within the 5kb window, the median length of ATAC-seq peaks is greater than 1kb (**Supp Fig 8D**). We have included a description of our motif analysis on (**Page 11; Lines 523,524**).

Page 10, Association of nascent transcription to epigenetic changes and contact frequency: Were the authors able to detect enhancer RNA expression with their start-seq dataset, and does eRNA expression correlate contact frequency or dominant loops?

We appreciate the reviewer's comment. In response, we conducted additional analyses to delve into the kinetic changes in enhancer RNA (eRNA) following acute androgen treatment. We examined the nascent Start-seq signal across AR-bound and AR-free enhancers looping to either upregulated or downregulated genes (**Fig 2**). Notably, eRNA transcription at AR-bound enhancers looped to AR-upregulated gene promoters (E+AR) behaved similarly to AR binding, peaking at 4h and then decreasing at 16h and 72h. In contrast, AR-bound enhancers looped to promoters of AR-downregulated genes (E+dAR) did not exhibit such dynamics. As this was not observed with histone modification, this may suggest a functional role of eRNAs in AR-mediated gene expression regulation. Interestingly, the peak in eRNA transcription at AR-bound enhancers preceded changes in accessibility and chromatin loop contact frequency. Furthermore, our new epigenetic correlation of promoters and enhancers (**Supp Fig 13**) displays a stronger correlation between AR binding and nascent RNA expression in enhancers

compared to promoters. This suggests that AR may play a role in inducing nascent RNA expression specifically within enhancer regions. We have included these results and discussed them further in our manuscript (**Pages 9, 12; Lines 429-436, 591-594**).

Line 499-500: CRE-promoter interactions of androgen-regulated genes (n=88) but its 89 in Supp Fig 1 A,B. Also, I couldn't find a Supp Table 2. Is it labelled as Supp Table 1a in the attached files?

We apologize for the ambiguity in gene numbers. Among the 89 accessible promoter regions of AR-regulated genes, 88 exclusively formed loops with another CRE. The discrepancy occurred as we initially defined the AR-regulated genes from HALLMARKS_ANDROGEN_RESPONSE before stratifying based on the HiChIP data. Consequently, earlier figures indicate 89, while the subsequent ones correctly reflect 88. We have corrected this throughout our figures (**Supp Fig 1**). Further, **Supp Table 2** now provides information on whether the neighbors of a given AR-upregulated gene form a dominant loop at various time points during androgen regulation.

Reviewer #3

It is crucial for the authors to address the gap in knowledge regarding whether the proposed multi-enhancer model can account for transcriptional regulation of AR independent of cell types. Given that the authors have solely utilized LNCaP cells for deriving the bulk sequencing data, it prompts the question of whether this finding is unique to cancer cells or if it extends to other cell types including non-cancerous prostate cells.

We would like to thank the reviewer for their feedback on the translation of the multi-enhancer model. Although our study primarily centered on AR-mediated transcription, we observed a similar multi-enhancer phenotype across the entire transcriptome (**Fig 5 and Supp Fig 9; previously Supp Fig 7**). These findings suggest that the proposed model does not exclusively occur only with AR. Supporting this hypothesis, a similar trend has been observed for other transcription factors (**PMID: 37989525, 30031775**). However, we acknowledge the limitation of this study and understand the necessity for additional exploration using different cell types, including those that are AR-independent or non-cancerous. We have discussed this limitation in our manuscript (**Pages 13, 14; Lines 640-642, 662-666**). Overall, we hope that our research will lead to further studies on transcription regulation through multiple chromatin interactions, thereby guiding future research toward a broader understanding of gene expression.

What accounts for the differential H3K27ac levels in AR-bound enhancer regions compared to AR-regulated promoter regions (Figure 2) and its potential impact on AR-regulated gene transcription? Some studies suggest that H3K27ac does not significantly affect the transcriptome (PMID: 32085783), while others indicate crosstalk between H3K27ac and H3K4me3 (<https://doi.org/10.1038/s41598-021-95398-5>). I hope the authors can further contribute to this discussion.

We thank the reviewer for raising this important point. As we observed changes in both upregulated and downregulated genes, our findings suggest that AR binding triggers an increase in H3K27ac at cis-regulatory elements (CREs) that is independent of steady-state gene expression (Fig 2) and maximal nascent mRNA expression (Fig 4b). This suggests that the enhancer-associated histone marks are driven by transcription factor binding and do not directly cause gene expression. Instead, from this kinetic multiomic dataset we found that AR-driven gene expression primarily correlates to the changes in contact frequency (Fig 3). The independent regulation of H3K27ac in our work aligns with published work (PMID: 32085783). At promoters, we noticed a slight decrease in the H3K27ac signal across all promoters. This decrease does not appear to be driven by AR, as it was also observed in AR-independent promoters (Fig 2). This could imply an indirect mechanism potentially driven by competition for histone acetyltransferases within the nucleus (PMID: 21321607). We did not observe AR-driven crosstalk between H3K27ac and H3K4me3 (Fig 2, Fig 3F). We have expanded our discussion of this phenotype in our manuscript (Page 12; Lines 586-591).

No validations were conducted to confirm any of the AR regulated gene expression (downregulated and upregulated) shown in the bulk data throughout the study.

We value the reviewer's insights regarding the validity of gene expression. First, we confirmed the changes in the gene expression profile (Fig 1E, Fig4A, Supp Fig 4A; previously Supp Fig 3A). We have added new figures to further clarify our rationale including a z-score heatmap (Supp Fig 1C, D). This heatmap demonstrates that these genes exhibit upregulation upon androgen treatment, with peak expression occurring at specific time points following the initial observation. For downregulation, we defined a particular time point -16h- since this exhibited the most pronounced regulatory response based on the contact frequency change results from the AR-upregulated genes. To support this, we showed the z-score heatmap for 16h downregulated genes (Supp Fig 4C, D), which indicates that most of these genes experience downregulation at 16h compared to other time points. Second, we would like to emphasize that our study employed not just RNA-seq, but also Start-seq as an orthogonal method for expression (PMID: 29378787). It's also worth noting that the precision of RNA-seq surpasses that of qPCR, typically eliminating the need for further validation (PMID: 33665610).

All the bulk datasets used in the study are derived from the prostate cancer cell line LNCaP. It is important to note that AR in LNCaP harbors a missense mutation (AR-T878A), resulting in broad and promiscuous steroid and anti-steroid binding specificity (PMID: 1562539 and PMID: 15657128). Will the findings in this study change accounting for this mutation and by the fact that AR in LNCaP can be activated by other ligands other than testosterone or DHT.

We thank the reviewer for highlighting the potential impact of the T878A LBD AR mutation that is found in LNCaP cells (PMID: 15657128). While this mutation does impact ligand interactions, published work suggests that T878A mutation only minimally impacts the protein structure when bound to androgen. Therefore both wild-type and T878A variant AR are likely to have a very similar phenotype when treated with agonist. This is clearly observed with the similar AR

cistromes observed between LNCaP and primary PCa clinical samples (PMID: 26813233) and expression of androgen-regulated genes like *KLK3/PSA* (PMID: 27196756). Consequently, we maintain confidence that our results remain unaffected by this AR mutation. However, we have highlighted the T878A mutation in our discussion (Page 13; Lines 642-644).

There are known androgen receptor variants (AR-V) with truncated ligand-binding domain that are constitutively active. The findings presented in this manuscript correspond to steroid induced AR (assuming androgens induce AR homodimerization) transcriptional regulations. How will the model presented in the study account for AR-V regulated transcription specifically in castration condition.

We appreciate the reviewer's interesting question. Our study delves into acute androgen-induced gene expression, specifically regulated by AR homodimers in a cell-line model (LNCaP) that does not express AR variants. Given the distinct interactome and transcriptome of AR-V compared to AR-FL (PMID: 30270106), there is some uncertainty about how our findings could extend to AR-V. However, it's crucial to emphasize that our initial multi-enhancer modeling was conducted in an AR-agnostic context. This suggests that the proposed dominance model would likely remain unchanged for AR-V. Furthermore, our observation of dominant enhancers at various time points (Supp Fig 10; previously Supp Fig 8) implies that their dominance persists despite androgen treatment. This suggests that dominant enhancers may not be defined by AR or AR-V, but rather could be pre-established prior to transcription factor binding.

Specifically in castration condition, we observe copy number gains in AR gene (which can result in the formation of AR gene containing extrachromosomal DNA) and copy number gains in enhancer regions (<https://doi.org/10.1038/s41467-023-40315-9>; PMID: 29909987). How will one apply the multi-enhancer model in this case scenario.

This is a very interesting question with significant clinical implications as the AR remains the main driver of castrate-resistant prostate cancer (mCRPC) (PMID: 37636316). Extrachromosomal, circular DNA (ecDNA) expressing AR is emerging as a common oncogenic alteration in prostate cancer genomes (PMID: 33836152). These "mobile enhancers" are in contact with chromosomal DNA and increase AR transcription. If we assume that selective pressure would maintain AR expression in mCRPC, it is possible that the dominant CREs-AR interaction would be selectively preserved in extrachromosomal DNA. Yet, it is also likely that there could be a massive rewiring of enhancer-promoter interactions with ecDNA that would change the specific enhancer driving expression. A recent preprint (PMID: 38712075) using droplet Hi-C found that ecDNA broadly interacts across the genome. Therefore it is likely that the AR looping structure would be dramatically changed leading to new promoter-enhancer interactions or enhancer "hijacking". Single-cell chromosomal confirmation data would be helpful to verify if the same AR CREs continue to dominate gene expression in the newly formed ecDNA. We have raised the potential implications of our multi-enhancer model to AR amplification in our discussion (Page 13; Lines 635,636).

In the article 'Multi-enhancer transcriptional hubs confer phenotypic robustness,' the authors suggest the existence of multi-enhancer hubs that facilitate phenotypic robustness under stress. I would like to raise the question to the authors: Does the multi-enhancer dominant model facilitate phenotypic robustness, and does it have an impact on the development of castration resistance?

We thank the reviewer for highlighting the potential impact of phenotypic robustness in the context of our multi-enhancer model. This intriguing paper (PMID: 31294690) proposes the existence of a critical threshold below which a gene fails to manifest its phenotypic effect. This phenotype demonstrates resilience to fluctuations in gene expression above the threshold -under ideal conditions- to genetic perturbations (i.e. deleting the gene's enhancer), suggesting a buffering strategy against stress. To explore phenotypic robustness, we investigated the number of loops (PMID: 36215037) for essential and non-essential genes in LNCaP compared to non-prostatic blood cell lines, GM12878, THP-1, and K562, according to the latest DepMap dataset (PMID: 29083409). The LNCaP-specific essential genes (n=265; colored) and non-essential genes (n=9541; grayed) were determined according to Gene Dependency scores (essentials: LNCaP > 0.5 & GM12878 < 0.5 & THP-1 < 0.5 & K562 < 0.5, non-essentials: LNCaP < 0.1 & GM12878 < 0.1 & THP-1 < 0.1 & K562 < 0.1). We compared the fold enrichment of the median number of loops in these cells, noting that only THP-1 had available H3K4me3 data. Although the median number of loop fold enrichment for H3K4me3 in LNCaP cells did not differ from THP-1 cells (green), for H3K27ac (blue), LNCaP-specific essential genes generally exhibited a higher number of loops compared to non-essential genes within LNCaP cells. This enrichment of loops in LNCaP-specific essential genes, compared to non-essential genes, was not as pronounced in blood cells. These findings support the hypothesis that a higher number of regulatory elements in cell-specific essential genes may contribute to phenotypic robustness.

Our CRISPRi experiments (**Fig 6A, Supp Fig 11B; previously Supp Fig 9B**) also offer insight into how the concept of enhancer dominance could be incorporated with this model. We observed that while dominant enhancers cannot completely abolish gene expression, their reduction results in a substantially greater decrease compared to other enhancers. The presence of multiple enhancers with varying regulatory strengths suggests a potential strategy for genes, allowing for heterogeneous regulation across different cellular contexts through the activation of different enhancer combinations. This supports the concept that having multiple enhancers with varying strengths allows genes to buffer against genetic or environmental perturbations. We have discussed the concept of phenotype robustness in our manuscript (**Page 13; Lines 631-635**).

Reviewer #4

The observed increase in looping frequency primarily relies on H3K27ac HiChIP data. However, upon androgen treatment, there's a notable increase in H3K27ac levels at enhancers of AR-activated genes (as shown in Fig.2). This increase could potentially cause bias on chromatin loop calculation. Despite the authors' argument to the contrary, using H3K4me3 as a comparison may not be appropriate since its ChIP-seq signals remain static under androgen treatment. The potential bias could impact the study's major conclusion.

We thank the reviewer for bringing up this point. We acknowledge the potential bias due to the immunoprecipitation of the target protein or histone modification. However, we would argue that this bias is minimal and does not impact our conclusions due to the following reasons:

1. Our study observed the same phenotype in HiChIP contact frequency with two distinct histone modifications, H3K27ac and H3K4me3. While there were changes in H3K27ac levels, H3K4me3 was not affected by androgen treatment. As the observed change in chromatin loop interactions was consistent across both histone marks (**Fig 3C, D**), this suggests that the effect that we observed did not solely depend on the enrichment of epigenetic modifications.
2. An increase in H3K27ac was observed at all AR-bound enhancers regardless of whether they looped to upregulated or downregulated genes (**Fig 2**). If our results were solely an artifact stemming from this bias, we would expect to observe a similar increase in contact frequency for downregulated genes. However, we did not observe a comparable increase in contact frequency increase for these loops of AR-downregulated genes as we did for loops of AR-upregulated genes (**Fig 3E**).
3. Calculations of contact frequency changes were conducted relative to randomly selected similarly expressing background genes within the same time point of DHT treatment. This approach effectively establishes a reference point within the experiment, thereby minimizing experimental biases across different time point experiments.

4. We have included additional analysis of the bias on HiChIP vs ChIP-seq signaling by calculating the change in both HiChIP and ChIP-seq on P+AR (Promoters of AR-upregulated genes) and E+AR (AR-bound enhancers of AR-upregulated Genes) (**Supp Fig 14**). This analysis was done with the assumption that a significant bias in immunoprecipitation would cause a correlation between the standard deviation (change) in both ChIP and HiChIP. However, this was not observed at either the H3K27ac promoter and enhancers (left panel) or H3K4me3 promoters (right panel).

We believe that this evidence demonstrates that changes in H3K27ac do not drive the increase in contact frequency.

The research's scope is quite limited, focusing solely on a single AR-positive prostate cancer cell line without addressing any physiological relevance of these findings. Even at the cellular level, questions remain about whether these mechanisms are broadly applicable across other steroid receptors or are simply specific to AR.

While this work focuses on androgen receptor (AR) activation, we propose that the multi-enhancer model is broadly applicable to other transcription factors. This is evident in non-AR-regulated genes, which show a similar dominant phenotype in chromatin looping (**Fig 5B, Supp Fig 9; previously Supp Fig 7**). We chose to characterize AR because it is selectively activated by external stimuli, allowing us to study how transcription factor binding changes locus-specific epigenetic and chromatin looping. Additionally, AR plays a pivotal role in prostate cancer proliferation, migration, invasion, and differentiation (**PMID: 15082523**). Since nearly all prostate cancers depend on AR, inhibiting AR-mediated transcription is the standard care for recurrent or metastatic disease. Although our study focused on this nuclear receptor, several of the individual phenotypes observed have also been seen with the estrogen receptor (**PMID: 37989525**) and glucocorticoid receptor (**PMID: 30031775**). We further validated these results with CRISPRi data, showing that these dominant loops contribute to transcription. Therefore, we believe that our findings broadly applicable to other transcription factors

The AR-regulated, AR-activated, AR-downregulated, and AR-independent genes need be clearly defined and be consistent from figures to figures.

We appreciate the reviewer's comment. In response, we have made revisions to clarify and ensure consistency throughout the paper (**Pages 6, 7, 9**). We have further generated figures displaying strong upregulation following androgen treatment (**Supp Fig 1C, D**) and downregulation at a specific time point, 16h (**Supp Fig 4C, D**).